# Signature-Informed Transformer for Asset Allocation

**Yoontae Hwang**[1]   **Stefan Zohren**[2]

## Abstract

Modern deep learning for asset allocation typically separates forecasting from optimization. We argue this creates a fundamental mismatch where minimizing prediction errors fails to yield robust portfolios. We propose the Signature Informed Transformer to address this by unifying feature extraction and decision making into a single policy. Our model employs path signatures to encode complex path dependencies and introduces a specialized attention mechanism that targets geometric asset relationships. By directly minimizing the Conditional Value at Risk we ensure the training objective aligns with financial goals. We prove that our attention module rigorously amplifies signature derived signals. Experiments across diverse equity universes show our approach significantly outperforms both traditional strategies and advanced forecasting baselines. The code is available at: https://github.com/Yoontae6719/Signature-Informed-Transformer-For-Asset-Allocation

## 1. Introduction

A central challenge in modern quantitative finance is strategic asset allocation: the dynamic construction of portfolios that are robust to the complex, non-linear behavior of financial markets (Markowitz, 1952). While foundational theories provided a basis for optimization, their assumptions of static correlations and normally distributed returns are often not adequate for navigating the non-stationary and path-dependent nature of today's markets (Cont, 2001; Fama, 1970). Deep learning offers a powerful toolkit to address these complexities, yet developing policies that yield stable, real-world performance remains a formidable task.

The predominant deep learning paradigm for this problem,

[1]Pusan National University, Busan, Republic of Korea
[2]University of Oxford, Oxford, United Kingdom. Correspondence to: Stefan Zohren <stefan.zohren@eng.ox.ac.uk>.

*Proceedings of the 43rd International Conference on Machine Learning*, Seoul, South Korea. PMLR 306, 2026. Copyright 2026 by the author(s).

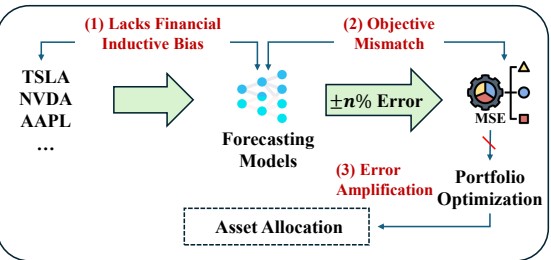

*Figure 1.* A depiction of flawed strategies for asset allocation

illustrated in Figure 1, is a decoupled, two-stage pipeline: a forecasting model first predicts asset returns, and these predictions are then fed into a downstream portfolio optimizer (Moody & Saffell, 2001). This approach has drawbacks and suffers from two critical issues. First, the forecasting models typically employed are general-purpose architectures. They lack the financial inductive biases necessary to model the idiosyncratic structures of financial markets, such as the intricate lead-lag relationships between assets. Without a model architecture that explicitly reflects market dynamics, such models struggle to distinguish genuine signals from noise. Second, and more critically, this pipeline creates an objective mismatch that leads to error amplification. The forecaster is trained to minimize a statistical metric like the Mean Squared Error (MSE), i.e. the average squared difference between estimated and actual values. This objective is agnostic to the downstream task of portfolio construction, where even minuscule prediction errors can be magnified by the optimizer into volatile and impractical portfolio weights. Furthermore, an MSE objective implicitly incentivizes the model to favor assets that are easier to predict, potentially not considering harder-to-predict assets with larger estimation errors and distorting the final allocation. We argue that a robust solution requires moving beyond this fragile pipeline. The challenge is to develop a unified policy that learns an end-to-end mapping from market data directly to portfolio weights while being architecturally designed to model the known geometric properties of financial time series (Buehler et al., 2019; Hwang et al., 2025a).

To this end, we introduce the **Signature-informed Transformer (SIT)**, a deep learning framework designed to learn robust, multi-asset allocation policies by directly addressing these challenges. SIT's contributions are unified within a synergistic architecture built on three pillars:

1. **Path-wise Feature Representation:** To better capture the complex dynamics of assets, the model generates features from each asset's price history using Rough Path Signatures. This technique offers a principled summary of a path's shape, encoding its trends and oscillations to provide a richer basis for decision-making (Lyons, 1998; Lyons & McLeod, 2022).

2. **Signature-Augmented Attention:** For modeling dependencies between assets, the model introduces a novel attention mechanism. It enhances attention scores with a term derived from the signature of asset pairs, which represents a robust measure of their lead-lag relationships (Bonnier et al., 2019). This allows the model to allocate attention based on geometric interactions, a crucial inductive bias for this problem.

3. **Decision Alignment:** To align the training process with the goal, the model is optimized directly for the quality of the portfolio allocation. Instead of aiming for statistical forecasting accuracy, its parameters are trained to minimize the Conditional Value-at-Risk (CVaR) of the portfolio's loss distribution, bridging the gap often found in two-stage pipelines.

## 2. Methodology

This section introduces the Signature-Informed Transformer (SIT), a novel approach to risk-aware portfolio allocation (Figure 2). All relevant literature can be found in the Appendix A. After a brief overview of the problem and path signatures, we detail the model's core components.

### 2.1. Preliminaries

**Notations.** Let $0 = t_0 < t_1 < \cdots < t_n = T$ denote a sequence of discrete times over the horizon $[0, T]$. We consider $d$ assets traded in a financial market, with price $S_{t_i}^j(\omega)$ referring to the value of asset $j \in \{1, \ldots, d\}$ at time $t_i$ under a particular market scenario $\omega \in \Omega$. The set $\Omega$ encapsulates all possible market paths. For convenience, we define the continuous-time vector process $\mathbf{S}_u(\omega) = (S_u^1(\omega), \ldots, S_u^d(\omega)) \in \mathbb{R}^d$, understanding that its values at discrete times $\{t_i\}$ coincide with the observed data $\{\mathbf{S}_{t_i}\}$. In practice, $\mathbf{S}_u$ on each interval $[t_i, t_{i+1}]$ can be reconstructed by an appropriate interpolation. A parametric asset allocation strategy is denoted by $\theta \in \Theta$, where $\Theta$ is the set of all feasible parameter configurations. At each decision time $t_i$, the policy outputs a sequence of long-only, fully invested portfolio weight vectors for the next $K$ periods, $\{\mathbf{w}_{t_i}^{(k)}(\theta)\}_{k=1}^K \subset \mathbb{R}_+^d$, with $\sum_{j=1}^d w_{t_i}^{(k),j}(\theta) = 1$ for each $k$. We parameterize each $\mathbf{w}_{t_i}^{(k)}$ via a softmax over the $k$-step-ahead predicted returns, $\mathbf{w}_{t_i}^{(k)}(\theta) = \text{softmax}(\hat{\boldsymbol{\mu}}_{t_i}^{(k)}(\theta)/\tau)$, where $\hat{\boldsymbol{\mu}}_{t_i}^{1:K}(\theta) \in \mathbb{R}^{K \times d}$ stacks the predictions for $k = 1, \ldots, K$. Our objective is to learn $\theta$ so as to maximize cumulative trading gains, subject

to uncertainty in market behavior.

A key ingredient in our framework is the use of path signatures to capture high-order variations and cross-asset interactions in price trajectories. For a continuous path $\mathbf{X} : [s, t] \to \mathbb{R}^d$, the signature $\text{Sig}(\mathbf{X}_{[s,t]})$ lies in the tensor algebra $\oplus_{k=0}^{\infty} (\mathbb{R}^d)^{\otimes k}$. When truncated at level $M$, it becomes a finite-dimensional vector denoted as $\text{Sig}^M(\mathbf{X}_{[s,t]}) = (1, \int_s^t d\mathbf{X}_u, \int_s^t \int_s^u d\mathbf{X}_r \otimes d\mathbf{X}_u, \ldots)$. In our financial context, $\mathbf{X}$ corresponds to the price process $\mathbf{S}_t$. First-order signature terms capture net increments for each asset, while second-order terms encode signed areas, revealing non-trivial correlations and lead-lag effects. Unless otherwise stated, we set the signature truncation order to $M = 2$, which is the lowest order that captures signed-area and directional lead-lag information. Appendix I reports a sensitivity analysis over $M \in \{1, 2, 3\}$. For clarity, the key notations are provided in Appendix B.

**Proposition 2.1** (Strict Lead-Lag Implies Positive Second-Order Signature (cf. (Chevyrev & Kormilitzin, 2016))). *Let $\mathbf{X}_t = (X_t^1, X_t^2)$ for $t \in [0, T]$ satisfy a strict lead-lag structure of Definition C.1. Then the second-level signature cross-term*

$$\mathcal{A}(\mathbf{X}) = \int_0^T X_t^1 dX_t^2 - \int_0^T X_t^2 dX_t^1 \qquad (1)$$

*is strictly positive. In particular, $\mathcal{A}(\mathbf{X}) > 0$.*

*Proof.* See Appendix C.2. $\square$

**Problem Formulation.** We frame the task as a sequential decision-making problem under uncertainty. At each decision point $t_i$, the objective is to construct portfolios of $d$ assets for each of the next $K$ periods $[t_i, t_{i+1}], \ldots, [t_{i+K-1}, t_{i+K}]$. The information set available at time $t_i$, denoted $\mathcal{F}_{t_i}$, comprises three components: (i) for each asset $j$, a sequence of truncated path signatures $\{\text{Sig}^M(S_{[t_{i-H+k-1}, t_{i-H+k}]}^j)\}_{k=1}^H$ over a lookback window of $H$ time steps (ii) pairwise cross-signatures $\text{Sig}^M((S^j, S^l)_{[t_{i-H}, t_i]})$ for all asset pairs $(j, l)$, capturing lead-lag relationships over the entire window and (iii) a sequence of deterministic calendar feature vectors $\{\mathbf{v}_{t_{i-H+k}}\}_{k=1}^H$, where $\mathbf{v}_t \in \mathbb{R}^F$. Our model, parameterized by $\theta \in \Theta$, learns a mapping

$$g_\theta : \mathcal{F}_{t_i} \longmapsto \hat{\boldsymbol{\mu}}_{t_i}^{1:K}(\theta), \qquad \hat{\boldsymbol{\mu}}_{t_i}^{1:K}(\theta) \in \mathbb{R}^{K \times d}, \quad (2)$$

which yields $k$-step-ahead expected returns for $k = 1, \ldots, K$. Portfolio weights for step $k$ are then obtained via $\mathbf{w}_{t_i}^{(k)}(\theta) = \text{softmax}(\hat{\boldsymbol{\mu}}_{t_i}^{(k)}(\theta)/\tau) \in \mathbb{R}^d$, ensuring a long-only, fully invested allocation at each future step. Note that $\hat{\boldsymbol{\mu}}_{t_i}^{1:K}$ is not trained with a prediction loss. It acts as the logits of the allocation layer, and gradients flow only from the portfolio objective below. Let $\mathbf{r}_{t_{i+k}}$ be the vector of realized

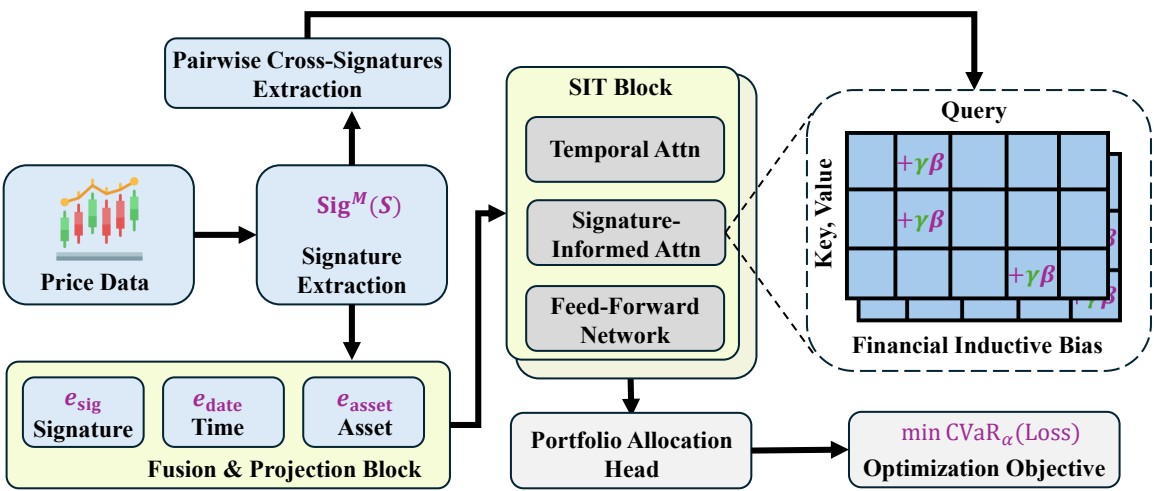

*Figure 2.* Overview of the Signature-Informed Transformer (SIT) architecture.

asset returns over $[t_{i+k-1}, t_{i+k}]$, and define the corresponding portfolio loss $L_{t_{i+k}}^{(k)}(\theta_\omega) = -(\mathbf{w}_{t_i}^{(k)}(\theta))^\top \mathbf{r}_{t_{i+k}}(\omega)$. The parameters $\theta$ are optimized by minimizing the expected Conditional Value-at-Risk (CVaR) of the $K$-step loss sequence within a scenario as

$$\min_{\theta \in \Theta} \; \mathbb{E}_{\omega \sim \mathcal{D}}[\mathrm{CVaR}_\alpha(\{L_{t_{i+k}}^{(k)}(\theta_\omega)\}_{k=1}^K)]. \quad (3)$$

A core assumption of this framework is that the complex, path-dependent market dynamics relevant for forecasting returns are effectively encoded within the signature features.

### 2.2. Signature-Informed Transformer (SIT)

**Signature Embeddings.** At a given decision time $t_i$, the initial representation for each asset $j$ and lookback slice $k \in \{1, \dots, H\}$ is constructed by fusing three distinct information sources. First, the truncated path signature of the asset's price history over the slice's interval, $\mathbf{s}_{k,j} = \mathrm{Sig}^M(S_{[t_{i-H+k-1}, t_{i-H+k}]}^j) \in \mathbb{R}^{d_{\mathrm{sig}}}$, is projected into the model's hidden space $\mathbb{R}^{d_{\mathrm{model}}}$ using a linear layer to form a path embedding $\mathbf{e}_{\mathrm{sig},k,j}$. Second, the vector of calendar features for that slice, $\mathbf{v}_{t_{i-H+k}} \in \mathbb{R}^F$, is projected to create a time embedding $\mathbf{e}_{\mathrm{date},k} \in \mathbb{R}^{d_{\mathrm{model}}}$, which is shared across all assets for slice $k$. Third, to encode unique, time-invariant characteristics, each asset $j \in \{1, \dots, d\}$ is assigned a learnable embedding vector $\mathbf{e}_{\mathrm{asset}}^j \in \mathbb{R}^{d_{\mathrm{model}}}$. These three embeddings are concatenated and passed through a final linear projection to produce the input token $\mathbf{x}_{k,j}$ for the first Transformer layer:

$$\mathbf{x}_{k,j} = W_{\mathrm{proj}}[\mathbf{e}_{\mathrm{sig},k,j} \oplus \mathbf{e}_{\mathrm{date},k} \oplus \mathbf{e}_{\mathrm{asset}}^j] \in \mathbb{R}^{d_{\mathrm{model}}} \quad (4)$$

where $\oplus$ denotes concatenation. The resulting input tensor for time $t_i$, of shape $H \times d \times d_{\mathrm{model}}$, encapsulates pathwise, temporal, and asset-specific information. Note that the signature update uses Chen relation (Lyons et al., 2007;

Moreno-Pino et al., 2024) and therefore avoids recomputation over the full history.

**Signature-Informed Self-Attention.** The core of the model's cross-asset reasoning lies in a novel attention mechanism that operates along the asset dimension, following a standard causal self-attention pass along the temporal dimension within each factorized layer. This Signature-Informed Self-Attention dynamically modifies the attention scores between pairs of assets based on their explicit relational features encoded by path signatures. Let the output of the temporal attention and its subsequent feed-forward network for a given layer be denoted by the tensor $\mathbf{X}' \in \mathbb{R}^{H \times d \times d_{\mathrm{model}}}$. For each time slice $k \in \{1, \dots, H\}$, we have a set of $d$ asset vectors $\{\mathbf{x}'_{k,1}, \dots, \mathbf{x}'_{k,d}\}$, where each $\mathbf{x}'_{k,j} \in \mathbb{R}^{d_{\mathrm{model}}}$. The asset-wise attention treats the time dimension as a batch dimension, processing $H$ independent attention calculations.

The mechanism is built upon a standard multi-head self-attention framework with $N_H$ heads. For a given time slice $k$, the collection of asset vectors $\mathbf{X}'_k = (\mathbf{x}'_{k,1}, \dots, \mathbf{x}'_{k,d})^\top \in \mathbb{R}^{d \times d_{\mathrm{model}}}$ is linearly projected to generate queries, keys, and values:

$$\mathbf{Q}_k = \mathbf{X}'_k W_Q \in \mathbb{R}^{d \times d_{\mathrm{model}}} \quad (5)$$

$$\mathbf{K}_k = \mathbf{X}'_k W_K \in \mathbb{R}^{d \times d_{\mathrm{model}}} \quad (6)$$

$$\mathbf{V}_k = \mathbf{X}'_k W_V \in \mathbb{R}^{d \times d_{\mathrm{model}}} \quad (7)$$

where $W_Q, W_K, W_V \in \mathbb{R}^{d_{\mathrm{model}} \times d_{\mathrm{model}}}$ are learnable weight matrices. These are then reshaped for multi-head computation, yielding per-head tensors $\mathbf{Q}_{k,h}, \mathbf{K}_{k,h}, \mathbf{V}_{k,h} \in \mathbb{R}^{d \times d_k}$ for each head $h \in \{1, \dots, N_H\}$, where $d_k = \frac{d_{\mathrm{model}}}{N_H}$. The innovation lies in the computation of an additive bias term.

This bias is a function of both pairwise relational characteristics and current asset states. The first component uses the cross-signature feature over the entire lookback window

$[t_{i-H}, t_i]$. For each pair of assets $(j, l)$, we denote the vector representation of this feature as $\mathbf{c}_{i,j,l} \in \mathbb{R}^{d_{\text{cross-sig}}}$. These features, encoding relational information for the pair $(j, l)$, are projected into a specialized embedding space using a dedicated MLP, denoted $\text{MLP}_\beta$, to produce a tensor of relational embeddings, $\boldsymbol{\beta}_{i,j,l}$:

$$\boldsymbol{\beta}_{i,j,l} = \text{MLP}_\beta(\mathbf{c}_{i,j,l}) \in \mathbb{R}^{N_H \times d_\beta} \qquad (8)$$

Here, $d_\beta$ is the bias embedding dimensionality, and a separate embedding is learned for each attention head. The second component introduces dynamism. The query asset's representation from the temporal stage, $\mathbf{x}'_{k,j}$, is used to generate a dynamic query vector via another MLP, $\mathbf{q}^{\text{dyn}}_{k,j} = \text{MLP}_q(\mathbf{x}'_{k,j}) \in \mathbb{R}^{N_H \times d_\beta}$. This vector $\mathbf{q}^{\text{dyn}}_{k,j}$ represents the *informational need* of asset $j$ at slice $k$. The dynamic attention bias, $b_{k,h,j,l}$, for each head $h$, query asset $j$, and key asset $l$ at time slice $k$, is computed via an inner product as

$$b_{k,h,j,l} = \langle (\mathbf{q}^{\text{dyn}}_{k,j})_h, (\boldsymbol{\beta}_{i,j,l})_h \rangle \qquad (9)$$

where $(\cdot)_h$ denotes the vector for head $h$. This forms a complete bias matrix $\mathbf{B}_k \in \mathbb{R}^{N_H \times d \times d}$ for each time slice $k$. This allows the model to selectively amplify or suppress attention based on whether a signature-encoded relationship is pertinent to the query asset's current state.

This dynamic bias matrix is scaled by a learnable, strictly positive scalar gate, $\gamma > 0$ (parameterized as $\gamma = \text{softplus}(\hat{\gamma})$), which controls the overall magnitude of the signature-based influence. The final attention logits are:

$$\text{Logits}_{k,h} = \frac{\mathbf{Q}_{k,h}\mathbf{K}^\top_{k,h}}{\sqrt{d_k}} + \gamma \mathbf{B}_{k,h} \in \mathbb{R}^{d \times d} \qquad (10)$$

The attention weights, $\boldsymbol{\alpha}_{k,h} \in \mathbb{R}^{d \times d}$, are obtained by applying the softmax function row-wise. The output for each head is computed by multiplying the attention weights with the value matrix.

**Proposition 2.2** (Positive directional derivative of attention weight). *Assume $d \geq 2$, $\gamma > 0$, and fix $(k, h, j, l)$. Let the query vector $(\mathbf{q}^{\text{dyn}}_{k,j})_h \in \mathbb{R}^{d_\beta}$ satisfy $\|(\mathbf{q}^{\text{dyn}}_{k,j})_h\|_2 > 0$. For*

$$z_{j,m} = \frac{(\mathbf{Q}_{k,h}\mathbf{K}^\top_{k,h})_{j,m}}{\sqrt{d_k}} + \gamma \langle (\mathbf{q}^{\text{dyn}}_{k,j})_h, (\boldsymbol{\beta}_{i,j,m})_h \rangle \quad (11)$$

$$\alpha_{j,m} = \frac{e^{z_{j,m}}}{\sum_{r=1}^d e^{z_{j,r}}}, \qquad (12)$$

*assume $0 < \alpha_{j,l} < 1$. Then the directional derivative of $\alpha_{j,l}$ with respect to $\boldsymbol{\beta}_{i,j,l}$ in the direction $(\mathbf{q}^{\text{dyn}}_{k,j})_h$ equals $D^{(\boldsymbol{\beta})}_{(\mathbf{q}^{\text{dyn}}_{k,j})_h} \alpha_{j,l} = \gamma \alpha_{j,l}(1 - \alpha_{j,l}) \|(\mathbf{q}^{\text{dyn}}_{k,j})_h\|^2_2 > 0$.*

*Proof.* See Appendix C.3. □

Intuitively, when a relational signature is aligned with a query asset's current informational need, strengthening that signature should raise the model's attention to the counterpart. Formally, Proposition 2.2 shows that, for fixed $\gamma > 0$, the directional derivative of $\alpha_{j,l}$ with respect to $(\boldsymbol{\beta}_{i,j,l})_h$ along $(\mathbf{q}^{\text{dyn}}_{k,j})_h$ is strictly positive, i.e., $D^{(\boldsymbol{\beta})}_{(\mathbf{q}^{\text{dyn}}_{k,j})_h} \alpha_{j,l} = \gamma \alpha_{j,l}(1 - \alpha_{j,l})\|(\mathbf{q}^{\text{dyn}}_{k,j})_h\|^2_2 > 0$. By contrast, the effect of increasing the gate $\gamma$ itself on $\alpha_{j,l}$ depends on alignment relative to other keys: $\frac{\partial \alpha_{j,l}}{\partial \gamma} = \alpha_{j,l}\left(b_{j,l} - \sum_{m=1}^d \alpha_{j,m} b_{j,m}\right)$, where $b_{j,m} = \langle (\mathbf{q}^{\text{dyn}}_{k,j})_h, (\boldsymbol{\beta}_{i,j,m})_h \rangle$. Thus, $\gamma > 0$ scales the influence of signature alignment, i.e. attention to pairs with above-average alignment increases as $\gamma$ grows, while attention to below-average alignment decreases.

Finally, the outputs from all heads are concatenated and passed through a final linear projection $W_O$, followed by a residual connection and layer normalization:

$$\text{Head}_{k,h} = \text{softmax}(\frac{\mathbf{Q}_{k,h}\mathbf{K}^\top_{k,h}}{\sqrt{d_k}} + \gamma \mathbf{B}_{k,h})\mathbf{V}_{k,h} \qquad (13)$$

$$\mathbf{O}_k = \text{Concat}(\text{Head}_{k,1}, \ldots, \text{Head}_{k,N_H})W_O \qquad (14)$$

$$\mathbf{X}''_k = \text{LayerNorm}(\mathbf{X}'_k + \text{Dropout}(\mathbf{O}_k)) \qquad (15)$$

The resulting collection $\{\mathbf{X}''_k\}_{k=1}^H$ is the output of the Signature-Informed Self-Attention block. The computational gain comes from factorizing attention over the temporal and asset dimensions: flattened attention over $H \times d$ tokens costs $\mathcal{O}(H^2 d^2 d_{\text{model}})$, whereas the factorized Transformer costs $\mathcal{O}(dH^2 d_{\text{model}} + Hd^2 d_{\text{model}})$. The signature-informed bias adds the pairwise cross-signature overhead; with $C_{\text{cross-sig}}(M)$ denoting the per-pair computation/projection cost, the total cost is $\mathcal{O}(dH^2 d_{\text{model}} + Hd^2 d_{\text{model}} + d^2 C_{\text{cross-sig}}(M))$.

**Training Strategy** The model is trained end-to-end to optimize portfolio performance under a risk-aware objective. The final output tensor from the Transformer stack, $\mathbf{X}'' \in \mathbb{R}^{H \times d \times d_{\text{model}}}$, summarizes pathwise and cross-asset information over the lookback window. An output head maps this representation at decision time $t_i$ to $K$-step-ahead return predictions: a linear projection (optionally preceded by pooling over the $H$ slices or using the last slice) produces $\hat{\boldsymbol{\mu}}^{1:K}_{t_i} \in \mathbb{R}^{K \times d}$. For each forecast step $k \in \{1, \ldots, K\}$, the predicted returns $\hat{\boldsymbol{\mu}}^{(k)}_{t_i} \in \mathbb{R}^d$ are converted into long-only portfolio weights via $\mathbf{w}^{(k)}_{t_i} = \text{softmax}(\hat{\boldsymbol{\mu}}^{(k)}_{t_i}/\tau)$, where $\tau > 0$ controls allocation concentration.

Let $\mathbf{r}_{t_{i+k}}$ denote the realized asset-return vector over $[t_{i+k-1}, t_{i+k}]$. The step-$k$ portfolio loss is $L^{(k)}_{t_{i+k}}(\theta_\omega) = -(\mathbf{w}^{(k)}_{t_i}(\theta))^\top \mathbf{r}_{t_{i+k}}(\omega)$. The overall objective is formally

stated as

$$\min_{\theta} \mathbb{E}_{\omega \sim \mathcal{D}}[\text{CVaR}_{\alpha}(\{L^{(k)}(\theta_{\omega})\}_{k=1}^{K})], \quad (16)$$

No auxiliary prediction losses are used. Eq. (16) is the sole training signal, avoiding the objective-mismatch issues discussed in Section A. For each scenario $\omega$, the inner $\text{CVaR}_{\alpha}$ is taken over the intra-scenario empirical distribution. The following derivation shows the dual form and its empirical counterpart used for optimization:

$$\mathcal{L}(\theta) = \mathbb{E}_{\omega \sim \mathcal{D}}[\text{CVaR}_{\alpha}(\{L^{(k)}(\theta_{\omega})\}_{k=1}^{K})] \quad (17)$$

$$= \mathbb{E}_{\omega \sim \mathcal{D}}\Big[\min_{\nu_{\omega} \in \mathbb{R}}\big(\nu_{\omega} + \frac{1}{(1-\alpha)K}\sum_{k=1}^{K}(L^{(k)}(\theta_{\omega}) - \nu_{\omega})^{+}\big)\Big] \quad (18)$$

$$\approx \frac{1}{N}\sum_{i=1}^{N}\min_{\nu_{i} \in \mathbb{R}}\big(\nu_{i} + \frac{1}{(1-\alpha)K}\sum_{k=1}^{K}(L^{(k)}(\theta_{\omega_{i}}) - \nu_{i})^{+}\big). \quad (19)$$

To incorporate risk aversion, we made the choice in Eq. (17) to minimizing the expected CVaR of the intra-scenario loss distribution, which is the objective in (16). Eq. (18) leverages the dual representation of CVaR (Rockafellar et al., 2000) under the confidence–level convention: $\text{CVaR}_{\alpha}(Z) = \min_{\nu \in \mathbb{R}}(\nu + \frac{1}{1-\alpha}\mathbb{E}[(Z - \nu)^{+}])$ with tail mass $1 - \alpha$. Thus $\nu_{\omega}$ equals the $\alpha$–quantile ($\text{VaR}_{\alpha}$) of the intra–scenario loss distribution. Finally, Eq. (19) presents the empirical objective function used in training, where the expectation $\mathbb{E}_{\omega \sim \mathcal{D}}$ is approximated by an average over a batch of $N$ scenarios $\{\omega_{i}\}_{i=1}^{N}$. For each scenario $\omega_{i}$, the optimal $\hat{\nu}_{i}$ is the empirical $\alpha$–quantile of its losses $\{L^{(k)}(\theta_{\omega_{i}})\}_{k=1}^{K}$.

## 3. Experiment

### 3.1. Implementation details

**Dataset** Experiments used three portfolios of 30, 40, and 50 S&P 100 companies. We also selected two additional portfolios of 10 and 20 assets from the DOW30 to validate performance against a different index composition, which is often characterized as more concentrated. Furthermore, to evaluate robustness across different market dynamics, we included two portfolios consisting of 50 and 100 assets from the CSI 300 index. The daily price data was sourced from Wharton Research Data Services (WRDS). The data was partitioned chronologically into distinct training, validation, and test periods. The training set spans from January 1, 2000, to December 31, 2016. The validation set from January 1, 2017, to December 31, 2019 and the test set from January 1, 2020, to December 27, 2024. This split covers multiple market regimes, including the recent volatility.

**Baseline Models** The performance of our proposed model, SIT, is compared against a comprehensive suite of bench-marks spanning traditional and deep learning approaches. Traditional baselines include **Equally Weighted Portfolio (EWP)** (DeMiguel et al., 2009), **Global Minimum Variance (GMV)** (Clarke et al., 2011; Markowitz, 1952), **Conditional Value-at-Risk (CVaR)** (Rockafellar et al., 2000) and **Hierarchical Risk Parity (HRP)** (Lopez de Prado, 2016). The portfolio optimization strategy forms the second stage of our deep learning-based comparisons, which use predictions from various state-of-the-art time-series fore-casting models as input. These forecasters include deep learning models such as **Autoformer** (Wu et al., 2021), **DLinear** (Zeng et al., 2023) **FEDformer** (Zhou et al., 2022), **PatchTST** (Nie et al., 2022), **iTransformer** (Liu et al., 2023), **Non-stationary Transformers (NSformer)** (Liu et al., 2022), **TimesNet** (Wu et al., 2022) and **RFormer** (Moreno-Pino et al., 2024). Details of the parameter search space are provided in Appendix D.

**Evaluation Metrics** The strategies were evaluated using four standard financial metrics, assuming a zero risk-free rate. Risk-adjusted performance was measured by the **Sharpe Ratio**, which accounts for total volatility, and the **Sortino Ratio**, a refinement that isolates downside risk by considering only downside deviation; higher values are superior for both. Overall growth was tracked by the **Final Wealth Factor**, while the **Maximum Drawdown** quantified the largest peak-to-trough percentage decline, with a lower value being preferable.

### 3.2. Can SIT deliver superior risk-adjusted performance?

We evaluate the out-of-sample asset allocations efficacy of our proposed model: SIT. The comprehensive performance metrics, including risk-adjusted returns and downside risk, are presented for the 40- and 50-asset universes (see Appendix 3.4 for the 30-asset universe experiment). Our analysis underscores that the quality of asset allocation, rather than raw predictive accuracy, is the decisive factor for success, a central tenet of our work. The empirical results, summarized in Table 1, demonstrate that SIT consistently and significantly outperforms all baseline models across the primary metrics of risk-adjusted return and wealth generation. In the 40-asset universe, for instance, SIT achieves a Sharpe Ratio of 0.6717 and a Sortino Ratio of 0.8232, decisively surpassing the next-best traditional baseline EWP and all deep learning counterparts. This translates into superior capital growth, with SIT yielding a Final Wealth Factor of 1.7903, the highest among all tested strategies.

The primary contribution of SIT becomes evident when contrasted with the predict-then-optimize models. These models, which rely on minimizing statistical forecasting error, exhibit poor and highly unstable portfolio performance. Many fail to outperform even simple heuristics. Their high

| *Panel A. Asset 40 Universe (S&P100)* | | | | |
| Models | Sharpe Ratio (↑) | Sortino Ratio (↑) | Maximum Drawdown (↓) | Final Wealth Factor (↑) |
|---|---|---|---|---|
| CVaR | 0.1531 | 0.2001 | 0.3516 | 1.0569 |
| EW | **0.5759** | **0.7153** | 0.3688 | **1.6439** |
| GMV | 0.4148 | 0.5337 | **0.2743** | 1.3258 |
| HRP | **0.4958** | 0.6171 | **0.3185** | 1.4561 |
| Autoformer | $0.2499 \pm 0.1405$ | $0.3423 \pm 0.1980$ | $0.3812 \pm 0.0480$ | $1.1809 \pm 0.2403$ |
| DLinear | $0.3167 \pm 0.1326$ | $0.4513 \pm 0.2005$ | $0.3621 \pm 0.0407$ | $1.2915 \pm 0.2133$ |
| FEDformer | $0.4006 \pm 0.2317$ | $0.5540 \pm 0.3192$ | $0.3647 \pm 0.0167$ | $1.5198 \pm 0.5703$ |
| iTransformer | $0.3157 \pm 0.0749$ | $0.4233 \pm 0.0943$ | $0.4136 \pm 0.0326$ | $1.2860 \pm 0.0147$ |
| NSformer | $0.4074 \pm 0.1151$ | $0.5820 \pm 0.1655$ | $0.4475 \pm 0.0672$ | $1.5129 \pm 0.3010$ |
| PatchTST | $0.3286 \pm 0.2021$ | $0.4540 \pm 0.2818$ | $0.4523 \pm 0.0838$ | $1.3409 \pm 0.3886$ |
| TimesNet | $0.3568 \pm 0.0782$ | $0.4959 \pm 0.1019$ | $0.4704 \pm 0.0701$ | $1.3765 \pm 0.1729$ |
| RFormer | $0.4901 \pm 0.1437$ | $\mathbf{0.6308 \pm 0.1828}$ | $\mathbf{0.3415 \pm 0.0482}$ | $\mathbf{1.5387 \pm 0.2353}$ |
| SIT (Ours) | $\mathbf{0.6717 \pm 0.0628}$ | $\mathbf{0.8232 \pm 0.0792}$ | $0.3611 \pm 0.0037$ | $\mathbf{1.7903 \pm 0.1023}$ |

| *Panel B. Asset 50 Universe (S&P100)* | | | | |
| Models | Sharpe Ratio (↑) | Sortino Ratio (↑) | Maximum Drawdown (↓) | Final Wealth Factor (↑) |
|---|---|---|---|---|
| CVaR | 0.2165 | 0.2858 | 0.3086 | 1.1170 |
| EW | **0.6008** | **0.7399** | 0.3604 | 1.6683 |
| GMV | 0.3947 | 0.4992 | **0.2678** | 1.2845 |
| HRP | 0.4637 | 0.5620 | **0.3258** | 1.4021 |
| Autoformer | $0.3899 \pm 0.1985$ | $0.5321 \pm 0.2870$ | $0.4356 \pm 0.1256$ | $1.4697 \pm 0.4573$ |
| DLinear | $0.2540 \pm 0.1215$ | $0.3557 \pm 0.1828$ | $0.3716 \pm 0.0193$ | $1.1883 \pm 0.1979$ |
| FEDformer | $0.4318 \pm 0.0692$ | $0.6039 \pm 0.1097$ | $0.4039 \pm 0.1143$ | $1.5286 \pm 0.1508$ |
| iTransformer | $0.5162 \pm 0.1367$ | $0.6761 \pm 0.1770$ | $0.4542 \pm 0.0239$ | $1.7910 \pm 0.3722$ |
| NSformer | $0.5238 \pm 0.0694$ | $\mathbf{0.7105 \pm 0.1033}$ | $0.4992 \pm 0.0975$ | $\mathbf{1.8138 \pm 0.1922}$ |
| PatchTST | $0.3821 \pm 0.1871$ | $0.5134 \pm 0.2635$ | $0.4255 \pm 0.1533$ | $1.4411 \pm 0.3814$ |
| TimesNet | $0.3050 \pm 0.3439$ | $0.4296 \pm 0.4864$ | $0.5181 \pm 0.1404$ | $1.3737 \pm 0.8857$ |
| RFormer | $\mathbf{0.5315 \pm 0.2519}$ | $0.6671 \pm 0.3255$ | $0.5202 \pm 0.0555$ | $\mathbf{1.8014 \pm 0.6303}$ |
| SIT (Ours) | $\mathbf{0.7715 \pm 0.0627}$ | $\mathbf{0.9743 \pm 0.0998}$ | $0.3271 \pm 0.0094$ | $\mathbf{1.9215 \pm 0.1792}$ |

*Table 1.* Portfolio performance of SIT versus baselines across 40- and 50-asset universes. The best, second-best, and third-best results for each metric are highlighted in red, blue, and **bold**, respectively. Consistent results on S&P100, Dow30, and CSI300 (up to 100 assets) are reported in Appendix G, with detailed parameter settings provided in Appendix D.

standard deviations across runs underscore the problem of error amplification, where small prediction inaccuracies are magnified by the downstream optimizer into fragile, impractical allocations. This finding empirically validates our core hypothesis. Optimizing for prediction is not a valid proxy for optimizing for allocation quality. In addition to its inductive biases designed for financial assets, SIT's decision-focused approach directly minimizes the portfolio's CVaR, fundamentally aligning the model's objective with the financial goal and thereby avoiding this critical pitfall. Furthermore, SIT demonstrates a superior risk-return profile compared to traditional quantitative strategies. While risk-minimizing models like Global Minimum Variance (GMV) achieve low Maximum Drawdown (MDD) (e.g., 0.2743 in the 40-asset case), they do so at the cost of substantially lower returns (Sharpe Ratio of 0.4148). SIT, conversely, maintains a competitive MDD (0.3611) while delivering significantly higher returns. For the results on the DOW30 and SCI300, please refer to Appendix 3.4.

### 3.3. Module-Level Contribution Experiments

To dissect the contribution of each architectural pillar of the **SIT**, we conduct a comprehensive ablation study. For this analysis, each ablated variant is created by independently removing a single key component from the full SIT model, while all other hyperparameters are held constant.

This module-drop protocol allows for a precise evaluation of each component's marginal contribution. The variants evaluated are: (i) **w/o CVaR**, which replaces the Conditional Value-at-Risk objective with a risk-neutral objective of maximizing mean returns (ii) **w/o Asset Attn**, which disables the entire Signature-Informed Self-Attention mechanism across assets (iii) **w/o Financial Bias**, which removes the signature-derived bias term from the attention scores, reverting to a standard self-attention mechanism and (iv) **w/o Gate** $\gamma$, which removes the learnable gate $\gamma$ that scales the financial bias.

| *Panel A. Asset 40 Universe* | | | | |
| Models | Sharpe | Sortino | MDD | Wealth |
|---|---|---|---|---|
| SIT (Ours) | 0.6717 | 0.8232 | 0.3611 | 1.7903 |
| w/o CVaR | $0.5691^c$ | $0.7057^b$ | 0.3695 | $1.6409^b$ |
| w/o Asset Attn | $0.5284^c$ | $0.6576^c$ | $0.3342^b$ | $1.5381^c$ |
| w/o Financial Bias | $0.6045^b$ | $0.7590^b$ | $0.3431^c$ | $1.6801^c$ |
| w/o Gate $\gamma$ | 0.5251 | $0.6489^c$ | $0.3470^b$ | $1.5470^c$ |

| *Panel B. Asset 50 Universe* | | | | |
| Models | Sharpe | Sortino | MDD | Wealth |
|---|---|---|---|---|
| SIT (Ours) | 0.7715 | 0.9743 | 0.3271 | 1.9215 |
| w/o CVaR | $0.5923^c$ | $0.7294^c$ | $0.3606^b$ | $1.6622^c$ |
| w/o Asset Attn | $0.6268^c$ | $0.7650^b$ | 0.3298 | $1.6562^b$ |
| w/o Financial Bias | $0.6047^c$ | $0.7545^b$ | 0.3224 | $1.6260^b$ |
| w/o Gate $\gamma$ | $0.5831^c$ | $0.7138^c$ | 0.3305 | $1.5945^c$ |

*Table 2.* Ablation study of SIT's core components. Superscripts $b$ and $c$ indicate statistical significance at $p < 0.05$ and $p < 0.001$ in paired tests against SIT (Ours).

The results, summarized in Table 2, underscore the impor-

tance of each design choice. The most critical element is the decision-focused approach. When the Conditional Value-at-Risk (CVaR) loss is replaced with a standard risk-neutral objective (w/o CVaR), the Sharpe Ratio on the 40-asset universe falls from 0.6717 to 0.5691. This demonstrates that direct optimization for risk-adjusted outcomes is essential for producing stable allocations that are resilient to tail events. The components of the Signature-Informed Self-Attention mechanism prove equally vital. Removing the asset-wise attention layer entirely (w/o Asset Attn) severely reduces the model's ability to reason about portfolio structure, causing a steep performance drop (Sharpe of 0.5284). Furthermore, removing just the signature-based inductive bias (w/o Financial Bias), i.e. reverting to a standard attention mechanism, still leads to significant degradation (Sharpe of 0.6045). This confirms that injecting principled geometric knowledge of lead-lag structures (Proposition 2.1) is more effective than forcing the model to learn these relationships from scratch. Finally, removing the learnable gate $\gamma$ (w/o Gate $\gamma$) is highly detrimental (Sharpe of 0.5251), highlighting that the model must learn to dynamically modulate the influence of these financial priors (Theorem 2.2) to adapt to changing market regimes.

### 3.4. Are signatures effective at driving attention?

SIT perturbs asset-axis attention logits by an additive, signature-induced bias, $\text{Logits}_{k,h} = \frac{\mathbf{Q}_{k,h}\mathbf{K}_{k,h}^\top}{\sqrt{d_k}} + \gamma\mathbf{B}_{k,h} \in \mathbb{R}^{d\times d}$ where $\mathbf{B}_{k,h}$ aggregates the alignment equation (9) between the query's informational need and the cross-signature embedding. Theorem 2.2 predicts that increasing this alignment in the query direction strictly raises attention weight on the corresponding key when $\gamma > 0$. Coupled with Proposition 2.1, which links persistent lead–lag to non zero second-order signatures, the method suggests a testable implication. Consequently, assets whose signatures are stronger should systematically attract more inbound attention. We formalize this implication by defining, at each decision time $t$, a per-asset signature-strength score $s_{t,j} = \frac{1}{HN_Hd}\sum_{k=1}^{H}\sum_{h=1}^{N_H}\sum_{m=1}^{d}(\mathbf{B}_{k,h})_{m,j}$ and the corresponding inbound-attention share $a_{t,j} = \frac{1}{HN_Hd}\sum_{k=1}^{H}\sum_{h=1}^{N_H}\sum_{m=1}^{d}\alpha_{k,h,m\to j}$. We then compute the Pearson correlation between $\{s_{t,j}\}_{j=1}^{d}$ and $(\{a_{t,j}\}_{j=1}^{d})$ at each $t$ and examine the distribution of these correlations across the test horizon.

In Figure 3, the distribution is right-skewed with means of 0.540 for the 40-asset universe and 0.611 for the 50-asset universe, indicating that stronger signature signals are associated with higher inbound attention mass. The heavy right tail shows frequent periods in which attention concentrates on assets whose signature-derived relations are most salient, while the paucity of negative correlations rules out a degeneracy in which the bias is ignored by the attention

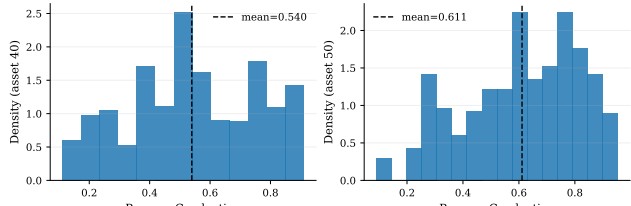

*Figure 3.* Distribution of correlations between signature strength and attention weights.

mechanism. These empirical patterns directly instantiate the monotonicity predicted by Theorem 2.2. So, as alignment $b_{k,h,j,l}$ strengthens, the induced change in $\alpha_{j,l}$ is positive, and the learned $\gamma$ scales this effect without flipping its sign. The financial significance of this correlation is

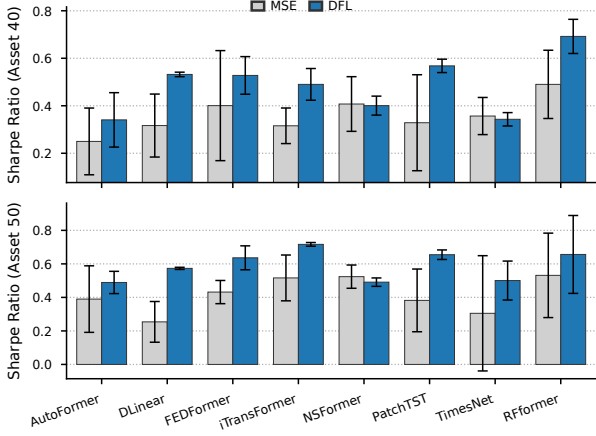

*Figure 4.* Out-of-sample Sharpe ratios after CVaR portfolio optimization on 40- and 50-asset S&P100 universes. Detailed numerical results for the 50-asset universe, including Sortino ratio, maximum drawdown, and terminal wealth, are provided in Appendix J.

twofold. First, $\mathbf{B}$ is not a static embedding. This couples dynamic queries $\mathbf{q}_{k,j}^{\text{dyn}}$ with cross-signatures $\boldsymbol{\beta}_{i,j,l}$. Hence, the correlation becomes most apparent when the model routes information toward assets whose relational signatures are currently informative. Second, because SIT is trained solely through the portfolio-level $\text{CVaR}_\alpha$ loss, the learned attention must improve tail-aware allocations rather than just forecast error. The observed right-skew therefore indicates that signatures are not merely present but actively utilized to amplify risk-relevant dependencies. In Appendix , Figure 7 overlays the learned gate $\gamma$ on portfolio drawdowns. We observe that higher values of $\gamma$ tend to cluster during volatile episodes. This suggests that SIT increases the weight of signature-based priors precisely when cross-asset relations are most informative and prediction noise is elevated. We therefore conjecture that this financial bias offers a plausible explanation for the results in Table 1. Specifically, it allows SIT to achieve the high risk-adjusted returns while

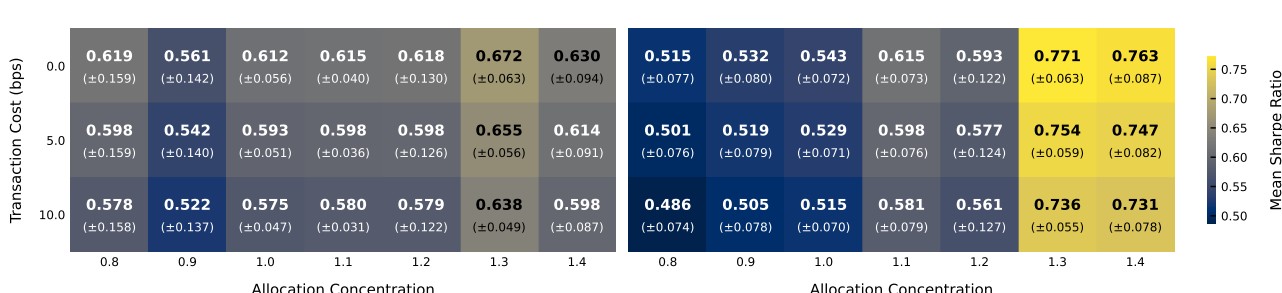

*Figure 5.* Sharpe ratio sensitivity to transaction costs and allocation concentration ($\tau$). Left: 40 assets Right: 50 assets.

maintaining a robust diversification profile comparable to the EWP.

### 3.5. Why PFL approach fail financial objectives?

Across all eight forecasting models, prediction alone does not translate into superior trading performance. As illustrated in Figure 4, which reports out-of-sample Sharpe ratios after CVaR optimization, the decision-focused learning (blue) consistently dominates the prediction-only approach (gray) across both the 40- and 50-asset universes. Moreover, as observed in Figure 1, the gap widens in the 50-asset universe, indicating that higher dimensionality amplifies the failure of the predict-then-optimize pipeline.

The failure of MSE-based approaches stems from the objective mismatch. See Appendix F for details on the two-stage implementation. Prediction-focused training optimizes a surrogate that is misaligned with the financial goal. Therefore, prediction losses weight all errors equally and are blind to the downstream mapping from forecasts to actions. A model that is optimal for $L_{\text{pred}}$ need not be even approximately optimal for CVaR. Consequently, tiny cross-sectional ranking errors induced by MSE training can be amplified by the optimizer, effectively flipping the identity of the largest weights. The evidence in Figure 4 and the mechanisms above explain why predict-then-optimize pipelines produce low and unstable Sharpe despite competitive $L_{\text{pred}}$. By differentiating through the portfolio layer and optimizing the risk metric of interest, decision-focused learning reshapes the logits so that only forecast features that improve allocation under constraints and tails are amplified. This alignment both raises risk-adjusted returns and tightens variability across runs, which we observe consistently across backbones and universes. We believe that aligning the training objective with the financial objective is necessary for turning predictive signals into reliable portfolios.

### 3.6. Sensitivity Analysis

We examine how proportional trading frictions and allocation concentration affect SIT. The concentration param-

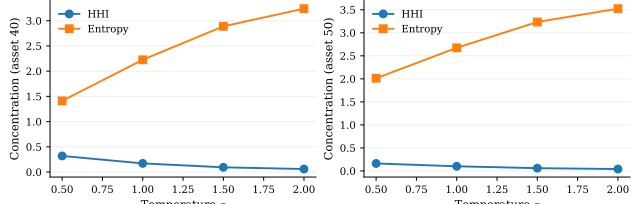

*Figure 6.* Impact of temperature $\tau$ on portfolio diversification.

eter $\tau$ is the softmax temperature in the allocation layer, $\mathbf{w}_t^{(k)} = \text{softmax}(\hat{\boldsymbol{\mu}}_t^{(k)}/\tau)$. Smaller $\tau$ concentrates capital, larger $\tau$ spreads it. We sweep $\tau \in \{0.8, ..., 1.4\}$. Transaction costs are one-way proportional fees of $c \in 0, 5, 10$ basis points per dollar traded. All other settings follow the main evaluation: long-only, fully invested, monthly ($k$-step) rebalancing on the 40- and 50-asset universes, and a zero risk-free rate for all risk-adjusted metrics. Figure 5 reports mean ($\pm$ std) Sharpe ratios for every $(\tau, c)$ pair, with the 40-asset universe on the left and 50-asset on the right. Two patterns are stable across universes. First, performance peaks at moderate dispersion, near $\tau \approx 1.3$. Second, frictions compress Sharpe roughly linearly over this range: moving from $0$ to $10$ bps reduces Sharpe by about $0.03$–$0.04$ at the optimum.

Trading costs predictably erode realized performance, yet the impact is mitigated when allocations avoid both extreme concentration (small $\tau$) and excessive diffusion (very large $\tau$). The interior optimum near $\tau \approx 1.3$ indicates that SIT's gains arise from robust allocation—balancing diversification with conviction rather than from raw prediction accuracy alone. The cost penalty is slightly smaller at the optimum in the 40-asset case (drop $0.034$) than for concentrated settings such as $\tau \in \{0.8, 0.9\}$ (drops $0.039$–$0.041$), whereas in the 50-asset universe the smallest penalty occurs at more concentrated $\tau$ (e.g., $\tau = 0.9$ drops $0.027$). This non-linearity suggests that the turnover induced by spreading capital interacts with cross-sectional breadth. With fewer assets, moderate diversification can temper trading; with more assets, broader participation slightly increases cost sensitivity.

Also, as shown in Figure 6, increasing the softmax temperature $\tau$ monotonically reduces cross-sectional concentra-

tion and increases diversification. Herfindahl–Hirschman Index $\mathrm{HHI}(\mathbf{w}) = \sum_j w_j^2$ declines, while Shannon entropy $\mathcal{H}(\mathbf{w}) = -\sum_j w_j \log w_j$ rises. For any fixed $\tau$, the 50-asset universe achieves lower HHI and higher $\mathcal{H}$ than the 40-asset universe, indicating a more diffuse allocation when the investable set is broader. These diagnostics provide an interpretable, one-to-one control of concentration through $\tau$, useful when desk policies cap the effective number of active lines or impose minimum diversification. We further examine the robustness of SIT to the CVaR confidence level $\alpha \in \{0.8, 0.9, 0.95, 0.99\}$. The detailed results are reported in Appendix J.1. Overall, larger $\alpha$ values lead to more conservative allocations with lower drawdowns, whereas smaller $\alpha$ values produce more aggressive wealth growth. The default choice $\alpha = 0.95$ provides the best risk-adjusted performance.

## 4. Conclusion

This work argues that effective quantitative portfolio management requires robust allocation policies, not just optimizing prediction accuracy. We introduce the Signature-informed Transformer, a novel framework using path signatures for rich feature representation, a signature-augmented attention mechanism for financial biases like lead-lag effects, and a training objective that directly minimizes portfolio Conditional Value-at-Risk. Our empirical results show that SIT decisively outperforms baselines, which often are harmed by objective mismatch and error amplification. SIT's performance remains superior under realistic transaction costs, underscoring the importance of its calibrated, signature-based architecture. While tested on U.S. equity data, this framework could be extended to higher-frequency, global, multi-asset markets. Ultimately, SIT provides a blueprint for ML systems to progress from forecasting towards a more end-to-end, risk-aware capital allocation.

## Impact Statement

This paper presents work whose goal is to advance the field of machine learning for financial asset allocation. There are many potential societal consequences of our work, none of which we feel must be specifically highlighted here.

## Funding

This work was supported by the National Research Foundation of Korea(NRF) grant funded by the Korea government(MSIT) (RS-2023-00242528).

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

# A. Appendix. Related works

**Deep Learning in Asset Allocation**    The application of deep learning in quantitative trading has largely bifurcated into two distinct paradigms. The first, the classic Predict Focused Learning (PFL) pipeline, focuses on developing return-prediction models. In this stream of research, complex architectures map market data to future price movements. For instance, Transformers have been adapted to capture temporal dependencies in asset prices for return forecasting (Fischer & Krauss, 2018; Yoo et al., 2021; Lim et al., 2021). Some models employ Graph Neural Networks (GNNs) to explicitly model inter-asset relationships, such as sector correlations, to improve prediction accuracy (Xu et al., 2021; Duan et al., 2025). Despite their architectural novelty, these methods inherit the fundamental flaws of a decoupled approach (Lee et al., 2024b). They suffer from objective mismatch, as optimizing for prediction error (e.g., Mean Squared Error) does not guarantee profitable portfolio construction, and are susceptible to error amplification, where small prediction inaccuracies lead to drastically suboptimal and unstable allocations (Chung et al., 2022). A more promising direction, which we term Decision Focused Learning (DFL), seeks to overcome these limitations by training policies end-to-end. These models learn a direct mapping from market state to portfolio allocations, optimizing a true financial objective like a risk-adjusted return metric. Foundational work demonstrated how to embed financial operators, such as portfolio value and Sharpe ratio, within a deep network, making the entire strategy differentiable and trainable via gradient descent (Buehler et al., 2019; Zhang et al., 2020; Costa & Iyengar, 2023). Recent research has increasingly emphasized embedding practical portfolio constraints into the model training phase. Typical examples include prohibiting short selling, ensuring full investment (i.e., portfolio weights sum to one), and placing upper or lower bounds on individual asset allocations, all of which are incorporated directly into the model architecture or loss function (Lee et al., 2024a; Hwang et al., 2025a). While these end-to-end frameworks efficiently align the model's training objective with financial goals, they often fall short in explicitly guiding the model to learn and utilize the diverse information present in multi-asset settings. This leaves a critical research gap. These models lack a strong financial inductive bias to explicitly represent the non-linear, path-dependent nature of price series and the geometric, time-local lead-lag relationships between assets. In our implementation, the predicted returns $\hat{\mu}$ serve only as internal logits for a differentiable allocation layer. All parameters are trained end-to-end solely through the portfolio-level CVaR objective, not a pointwise prediction loss, aligning with decision-focused learning. Our work addresses this gap by integrating the mathematical theory of path signatures directly into a transformer's attention mechanism, creating an optimization-aware model that is architecturally designed to understand the underlying geometry of market dynamics. See (Lee et al., 2024b; Hwang et al., 2025b) for more detailed review of asset allocations

**Transformer-Based Time Series Forecasting**    The success of the Transformer architecture in natural language processing has inspired its widespread adoption for time series forecasting. The core innovation, the self-attention mechanism, allows these models to dynamically weigh the importance of all past time steps when predicting future values, enabling them to capture complex, long-range dependencies without the sequential processing limitations of recurrent neural networks(Vaswani et al., 2017; Li et al., 2019). To extend the receptive field without incurring the quadratic cost of full attention, a stream of variants introduce sparsity or hierarchical structure. For example, LogSparse(Li et al., 2019), ProbSparse(Zhou et al., 2021) and related kernels discard low-magnitude query–key interactions to achieve $\mathcal{O}(L \log L)$ complexity while retaining global context. From a more fundamental time series data perspective, Autoformer(Wu et al., 2021), FEDformer(Zhou et al., 2022) and ETSformer(Woo et al., 2022) decompose signals into trend–seasonality (or frequency-domain) components so that long-horizon patterns can be modeled additively and multiplicatively with reduced error accumulation. More recent PatchTST(Nie et al., 2022) and TimesNet(Wu et al., 2022) patch neighboring observations or convolve multi-scale windows before attention, embedding stronger inductive biases for periodicity and aliasing control. While these innovations alleviate the long-range dependency bottleneck, they remain largely data-agnostic. When applied to financial series they struggle with regime-dependent non-stationary, heavy-tailed noise, and asynchronous cross-asset lead–lag effects, causing attention scores to lock onto transient outliers and degrading out-of-sample robustness (Cartea et al., 2023; Cont, 2001; Miori & Cucuringu, 2022). Our approach departs from this paradigm by embedding each asset's path in a Rough Path Signature space that is stable under time-reparameterization and robust to micro-structure noise, and by augmenting the attention logits with second-order cross-signature terms that encode the signed-area geometry underpinning lead–lag dynamics. Coupled with scenario-based optimization to hedge against structural breaks, SIT addresses both the generic long-range dependency problem and the finance-specific pathologies that limit existing Transformer forecasters.

**Path Signatures in Time Series and Finance**    The path signature, originating from Rough Path Theory, offers a non-parametric and faithful representation of streamed data by summarizing the geometry of a path as a sequence of iterated integrals (Lyons, 1998). A key property is its universality: any continuous function on the space of paths can be arbitrarily

well-approximated by a linear function of the signature's terms, making it a powerful basis for feature extraction (Chevyrev & Kormilitzin, 2016). In practice, the signature is truncated at a finite order $M$, yielding a vector $\text{Sig}^M(\mathbf{X})$ that is robust to irregular sampling due to its invariance to time reparameterization. However, this truncation introduces a trade-off, as the feature dimension grows exponentially with the order $M$ and polynomially with the path dimension $d$, posing a significant computational burden. This challenge has motivated alternatives like signature kernels, which compute inner products in the high-dimensional feature space implicitly, avoiding explicit feature construction (Király & Oberhauser, 2019). In machine learning, signatures provide a potent inductive bias for modeling systems with path-dependent memory. The most direct application involves using truncated signatures as static input features for standard models (Gyurkó et al., 2013). More sophisticated integrations are found in continuous-time models like Neural Controlled Differential Equations (CDEs), which learn a vector field that is controlled by the input path, effectively modeling the system's response to a driving signal (Kidger et al., 2020). For finance, a crucial insight arises from the signature's geometry: the second-order terms of a joint signature over two asset paths precisely encode their signed area, a direct and robust measure of their temporal lead-lag relationship (Lyons & McLeod, 2022). This property has been successfully leveraged to build kernels for detecting asymmetric dependencies between financial instruments, offering a principled alternative to traditional correlation measures (Bonnier et al., 2019). Recent advancements extend this to attention mechanisms. the Rough Transformer (Moreno-Pino et al., 2024) introduces multi-view signature attention to operate directly on continuous-time representations. Also, applications to finance span volatility/return modeling, derivatives, and market microstructure. Early studies extracted signature coordinates to forecast realized volatility and to detect temporal asymmetries (Gyurkó et al., 2013). In options, signatures parameterize no-arbitrage dynamics and enable data-driven pricing/hedging (Arribas et al., 2020), including transformer-style encoders fed with log/signatures (Tong et al., 2023). A crucial geometric motif is the second-order signed area,

$$A(X^i, X^j) = \int X^i \, dX^j - \int X^j \, dX^i, \tag{20}$$

which encodes temporal asymmetry and lead–lag; signature kernels exploit this to compare pairs or small baskets of assets (Chevyrev & Kormilitzin, 2016; Király & Oberhauser, 2019). Our architecture operationalization this motif at scale: SIT injects cross-asset signature information as a dynamic, query-conditioned bias inside attention, so that pairwise signed-area evidence modulates which assets attend to which others at each decision point (cf. proposition 2.1). While signatures mitigate non-stationarity and encode higher-order interactions, they incur truncation bias and can suffer from a curse of dimensionality as either degree $M$ or the number of assets grows; kernelization trades feature savings for quadratic kernel costs (Salvi et al., 2021; Bonnier et al., 2019). Compared with state-space or transformer baselines, signatures offer complementary bias—geometric invariances and lead–lag structure—rather than longer receptive fields alone. Prior signature-based works typically (i) use signatures as fixed inputs or kernels outside the attention mechanism and (ii) optimize predictive losses, not portfolio objectives (Gyurkó et al., 2013; Tong et al., 2023; Bonnier et al., 2019). SIT differs by coupling signature-augmented, cross-asset attention with end-to-end CVaR optimization for long-only, fully-invested portfolios, aligning representation, interaction, and objective (Buehler et al., 2019).

## B. Appendix. Notation

For clarity and ease of reference, Table 3 provides a comprehensive summary of the key notations used throughout this paper.

## C. Appendix. Mathematical Proofs

**Definition C.1.** (Strict Lead-Lag Structure) Let $\mathbf{X}_t = (X_t^1, X_t^2)$ be a continuous path of bounded variation on $[0, T]$. We say it possesses a *strict lead-lag structure* if there exist an integer $N \geq 1$ and a partition $0 = t_0 < t_1 < \cdots < t_{2N} = T$ of the interval $[0, T]$ such that the following conditions hold:

(i) For each $k \in \{0, 1, \ldots, N\}$, the coordinates coincide at the even-indexed partition points: $X_{t_{2k}}^1 = X_{t_{2k}}^2$. Let this common value be denoted by $S_k$.

(ii) For each $k \in \{1, 2, \ldots, N\}$:

- On $[t_{2k-2}, t_{2k-1}]$ (the $k$-th lead interval), $X_t^1$ varies to satisfy $X_{t_{2k-1}}^1 = S_k$, while $X_t^2$ remains constant at $S_{k-1}$.
- On $[t_{2k-1}, t_{2k}]$ (the $k$-th lag interval), $X_t^1$ remains constant at $S_k$, while $X_t^2$ varies to satisfy $X_{t_{2k}}^2 = S_k$.

| Symbol | Description | Type / Dimension |
|---|---|---|
| $\mathbb{R}$ | Set of real numbers | — |
| $\mathbb{E}[\cdot]$ | Expectation operator | — |
| $0 = t_0 < \cdots < t_n = T$ | Discrete decision times | Scalars |
| $d$ | Number of tradable assets | $\in \mathbb{N}$ |
| $S_{t_i}^j$ | Price of asset $j$ at time $t_i$ | Scalar |
| $\mathbf{S}_u$ | Price vector $(S_u^1, \ldots, S_u^d)$ | $\in \mathbb{R}^d$ |
| $\Omega$ | Set of market scenarios (price paths) | Sample space |
| $\theta \in \Theta$ | Trainable parameters / parameter space | Vector / set |
| $H$ | Look-back window length (time steps) | $\in \mathbb{N}$ |
| $K$ | Forecasting window length (time steps) | $\in \mathbb{N}$ |
| $M$ | Signature truncation level | $\in \mathbb{N}$ |
| $\mathrm{Sig}^M(\mathbf{X}_{[s,t]})$ | Truncated signature of path $\mathbf{X}$ up to level $M$ | $\in \mathbb{R}^{d_{\mathrm{sig}}}$ |
| $\mathrm{CVaR}_\alpha(\cdot)$ | Conditional Value-at-Risk at level $\alpha$ | Scalar |
| $\mathbf{s}_{k,j}$ | Signature vector for slice $k$, asset $j$ | $\in \mathbb{R}^{d_{\mathrm{sig}}}$ |
| $\mathbf{v}_t$ | Calendar/feature vector at time $t$ | $\in \mathbb{R}^F$ |
| $\mathbf{e}_{\mathrm{asset}}^j$ | Learnable embedding of asset $j$ | $\in \mathbb{R}^{d_{\mathrm{model}}}$ |
| $\mathbf{x}_{k,j}$ | Input token for slice $k$, asset $j$ | $\in \mathbb{R}^{d_{\mathrm{model}}}$ |
| $\mathbf{Q}, \mathbf{K}, \mathbf{V}$ | Query, key, value matrices (per slice) | $\in \mathbb{R}^{d \times d_{\mathrm{model}}}$ |
| $\boldsymbol{\beta}_{i,j,l}$ | Cross-signature embedding for pair $(j, l)$ | $\in \mathbb{R}^{N_H \times d_\beta}$ |
| $\mathbf{q}_{k,j}^{\mathrm{dyn}}$ | Dynamic query bias for asset $j$, slice $k$ | $\in \mathbb{R}^{N_H \times d_\beta}$ |
| $\gamma$ | Positive gate for signature bias | $\in \mathbb{R}_{>0}$ |
| $\{\mathbf{w}_{t_i}^{(k)}\}_{k=1}^K$ | Future portfolio weights at $t_i$ (long-only) | Each $\in \mathbb{R}^d$, $\sum w = 1$ |
| $\{\mathbf{r}_{t_{i+k}}\}_{k=1}^K$ | Realized returns for steps $1{:}K$ | Each $\in \mathbb{R}^d$ |
| $\{L_{t_{i+k}}^{(k)}\}_{k=1}^K$ | Portfolio losses for steps $1{:}K$ | Each scalar |
| $\hat{\boldsymbol{\mu}}_{t_i}^{1:K}$ | Predicted $k$-step-ahead returns for $k = 1, \ldots, K$ | $\in \mathbb{R}^K$ per asset; stacked as $\in \mathbb{R}^{K \times d}$ |
| $\tau$ | Softmax temperature (Allocation Concentration) | $\in \mathbb{R}_{>0}$ |

*Table 3.* Summary of the principal notation used throughout the paper.

(iii) For each $k \in \{1, 2, \ldots, N\}$, the change between synchronization points is non-zero, i.e., $S_k \neq S_{k-1}$.

**Proposition C.2.** *(Strict Lead-Lag Implies Positive Second-Order Signature) Let $\mathbf{X}_t = (X_t^1, X_t^2)$ for $t \in [0, T]$ satisfy the strict lead-lag structure of Definition C.1. Then the second-level signature cross-term*

$$\mathcal{A}(\mathbf{X}) = \int_0^T X_t^1 \, dX_t^2 - \int_0^T X_t^2 \, dX_t^1 \tag{21}$$

*is strictly positive. In particular, $\mathcal{A}(\mathbf{X}) > 0$.*

*Proof.* Let $\mathbf{X}_t = (X_t^1, X_t^2)_{t \in [0,T]}$ be a path of bounded variation with the strict lead-lag structure of Definition C.1. By this structure, there exists a partition $0 = t_0 < t_1 < \cdots < t_{2N} = T$ such that on each interval $[t_{2k-2}, t_{2k-1}]$ only $X^1$ varies (while $X^2$ remains constant), and on the following interval $[t_{2k-1}, t_{2k}]$ only $X^2$ varies (while $X^1$ is constant). Moreover, at the synchronization times $t_{2k}$ both coordinates coincide, and no increment is zero.

Recall from Definition C.1 the common values at the synchronization points:

$$S_{k-1} = X_{t_{2k-2}}^1 = X_{t_{2k-2}}^2 \quad \text{and} \quad S_k = X_{t_{2k}}^1 = X_{t_{2k}}^2. \tag{22}$$

Then $S_k \neq S_{k-1}$ by strictness. Let $\Delta S_k := S_k - S_{k-1}$. By construction, on $[t_{2k-2}, t_{2k-1}]$ (the $k$-th lead step) $X^1$ varies from $S_{k-1}$ to $S_k$ while $X^2$ stays at $S_{k-1}$; on $[t_{2k-1}, t_{2k}]$ (the lag step) $X^1$ remains $S_k$ while $X^2$ moves from $S_{k-1}$ to $S_k$.

Now we compute the cross-integral:

$$\mathcal{A}(\mathbf{X}) = \int_0^T X_t^1 \, dX_t^2 - \int_0^T X_t^2 \, dX_t^1. \tag{23}$$

Using the piecewise structure, we have for each $k$:

$$\int_{t_{2k-2}}^{t_{2k}} X_t^1 \, dX_t^2 = \int_{t_{2k-1}}^{t_{2k}} X_t^1 \, dX_t^2 \qquad \text{(since } dX_t^2 = 0 \text{ on } [t_{2k-2}, t_{2k-1}]) \tag{24}$$

$$= S_k \left[ X_{t_{2k}}^2 - X_{t_{2k-1}}^2 \right] \qquad \text{(since } X_t^1 = S_k \text{ is constant on } [t_{2k-1}, t_{2k}]) \tag{25}$$

$$= S_k \Delta S_k. \tag{26}$$

Similarly,

$$\int_{t_{2k-2}}^{t_{2k}} X_t^2 \, dX_t^1 = \int_{t_{2k-2}}^{t_{2k-1}} X_t^2 \, dX_t^1 \qquad \text{(since } dX_t^1 = 0 \text{ on } [t_{2k-1}, t_{2k}]) \tag{27}$$

$$= S_{k-1} \left[ X_{t_{2k-1}}^1 - X_{t_{2k-2}}^1 \right] \qquad \text{(since } X_t^2 = S_{k-1} \text{ is constant on } [t_{2k-2}, t_{2k-1}]) \tag{28}$$

$$= S_{k-1} \Delta S_k. \tag{29}$$

Summing over $k = 1$ to $N$ and subtracting:

$$\mathcal{A}(\mathbf{X}) = \sum_{k=1}^{N} (S_k \Delta S_k - S_{k-1} \Delta S_k) \tag{30}$$

$$= \sum_{k=1}^{N} (S_k - S_{k-1}) \Delta S_k \tag{31}$$

$$= \sum_{k=1}^{N} (\Delta S_k)^2. \tag{32}$$

Thus $\mathcal{A}(\mathbf{X}) = \sum_{k=1}^{N} (\Delta S_k)^2$. Since $S_k \neq S_{k-1}$ for each $k$ by condition (iii), we have $\Delta S_k \neq 0$, so each term $(\Delta S_k)^2$ is strictly positive. Therefore, the sum $\mathcal{A}(\mathbf{X})$ is strictly positive. $\qquad\square$

**Proposition C.3** (Positive directional derivative of attention weight). *Assume $d \geq 2$, $\gamma > 0$, and fix $(k, h, j, l)$. Let the query vector $(\mathbf{q}_{k,j}^{\text{dyn}})_h \in \mathbb{R}^{d_\beta}$ satisfy $\|(\mathbf{q}_{k,j}^{\text{dyn}})_h\|_2 > 0$. For*

$$z_{j,m} = \frac{(\mathbf{Q}_{k,h} \mathbf{K}_{k,h}^\top)_{j,m}}{\sqrt{d_k}} + \gamma \langle (\mathbf{q}_{k,j}^{\text{dyn}})_h, (\boldsymbol{\beta}_{i,j,m})_h \rangle, \qquad \alpha_{j,m} = \frac{e^{z_{j,m}}}{\sum_{r=1}^{d} e^{z_{j,r}}},$$

*assume $0 < \alpha_{j,l} < 1$. Then the directional derivative of $\alpha_{j,l}$ with respect to $\boldsymbol{\beta}_{i,j,l}$ in the direction $(\mathbf{q}_{k,j}^{\text{dyn}})_h$ equals*

$$D_{(\mathbf{q}_{k,j}^{\text{dyn}})_h}^{(\boldsymbol{\beta})} \alpha_{j,l} = \gamma \, \alpha_{j,l} \big(1 - \alpha_{j,l}\big) \left\| (\mathbf{q}_{k,j}^{\text{dyn}})_h \right\|_2^2 > 0. \tag{33}$$

*Proof.* For a fixed time slice $k$ and head $h$, the attention weight $\alpha_{k,h,j\to l}$ is the $l$-th component of the softmax function applied to the $j$-th row of the logits matrix. Let $z_{j,m}$ be the logit for query asset $j$ and key asset $m \in \{1, \ldots, d\}$.

$$z_{j,m} = \frac{(\mathbf{Q}_{k,h} \mathbf{K}_{k,h}^\top)_{j,m}}{\sqrt{d_k}} + \gamma b_{k,h,j,m} \tag{34}$$

The bias term $b_{k,h,j,l}$ is given by the inner product $b_{k,h,j,l} = \langle (\mathbf{q}_{k,j}^{\text{dyn}})_h, (\boldsymbol{\beta}_{i,j,l})_h \rangle$. The attention weight is:

$$\alpha_{k,h,j\to l} = \frac{\exp(z_{j,l})}{\sum_{m=1}^{d} \exp(z_{j,m})} \tag{35}$$

We wish to compute the directional derivative of $\alpha_{k,h,j\to l}$ with respect to the vector $(\boldsymbol{\beta}_{i,j,l})_h$ in the direction of $\mathbf{u} = (\mathbf{q}_{k,j}^{\text{dyn}})_h$, which is defined as $D_{\mathbf{u}} \alpha_{k,h,j\to l} = \langle \nabla_{(\boldsymbol{\beta}_{i,j,l})_h} \alpha_{k,h,j\to l}, \mathbf{u} \rangle$.

First, we find the gradient of $\alpha_{k,h,j \to l}$. By the chain rule,

$$\nabla_{(\boldsymbol{\beta}_{i,j,l})_h} \alpha_{k,h,j \to l} = \sum_{m=1}^{d} \frac{\partial \alpha_{k,h,j \to l}}{\partial z_{j,m}} \nabla_{(\boldsymbol{\beta}_{i,j,l})_h} z_{j,m} \tag{36}$$

The relational embedding $(\boldsymbol{\beta}_{i,j,l})_h$ only appears in the bias term $b_{k,h,j,l}$, and thus only affects the logit $z_{j,l}$. For any $m \neq l$, $\nabla_{(\boldsymbol{\beta}_{i,j,l})_h} z_{j,m} = \mathbf{0}$. Therefore, the sum collapses to a single term:

$$\nabla_{(\boldsymbol{\beta}_{i,j,l})_h} \alpha_{k,h,j \to l} = \frac{\partial \alpha_{k,h,j \to l}}{\partial z_{j,l}} \nabla_{(\boldsymbol{\beta}_{i,j,l})_h} z_{j,l} \tag{37}$$

The derivative of the softmax function is $\frac{\partial \alpha_{k,h,j \to l}}{\partial z_{j,l}} = \alpha_{k,h,j \to l}(1 - \alpha_{k,h,j \to l})$. The gradient of the logit $z_{j,l}$ with respect to $(\boldsymbol{\beta}_{i,j,l})_h$ is:

$$\nabla_{(\boldsymbol{\beta}_{i,j,l})_h} z_{j,l} = \nabla_{(\boldsymbol{\beta}_{i,j,l})_h} \left( \frac{(\mathbf{Q}_{k,h} \mathbf{K}_{k,h}^{\top})_{j,l}}{\sqrt{d_k}} + \gamma \langle (\mathbf{q}_{k,j}^{\text{dyn}})_h, (\boldsymbol{\beta}_{i,j,l})_h \rangle \right) = \gamma (\mathbf{q}_{k,j}^{\text{dyn}})_h \tag{38}$$

Substituting these back, we get the gradient of the attention weight:

$$\nabla_{(\boldsymbol{\beta}_{i,j,l})_h} \alpha_{k,h,j \to l} = \gamma \cdot \alpha_{k,h,j \to l}(1 - \alpha_{k,h,j \to l}) \cdot (\mathbf{q}_{k,j}^{\text{dyn}})_h \tag{39}$$

Now, we compute the directional derivative:

$$D_{(\mathbf{q}_{k,j}^{\text{dyn}})_h} \alpha_{k,h,j \to l} = \langle \gamma \cdot \alpha_{k,h,j \to l}(1 - \alpha_{k,h,j \to l}) \cdot (\mathbf{q}_{k,j}^{\text{dyn}})_h, (\mathbf{q}_{k,j}^{\text{dyn}})_h \rangle \tag{40}$$

$$= \gamma \cdot \alpha_{k,h,j \to l}(1 - \alpha_{k,h,j \to l}) \cdot \langle (\mathbf{q}_{k,j}^{\text{dyn}})_h, (\mathbf{q}_{k,j}^{\text{dyn}})_h \rangle \tag{41}$$

$$= \gamma \cdot \alpha_{k,h,j \to l}(1 - \alpha_{k,h,j \to l}) \cdot \|(\mathbf{q}_{k,j}^{\text{dyn}})_h\|^2 \tag{42}$$

By assumption, $\gamma > 0$. The attention weight satisfies $0 < \alpha_{k,h,j \to l} < 1$ (for any non-degenerate case with at least two assets), so the term $\alpha_{k,h,j \to l}(1 - \alpha_{k,h,j \to l})$ is strictly positive. By assumption, $(\mathbf{q}_{k,j}^{\text{dyn}})_h \neq \mathbf{0}$, so its squared norm $\|(\mathbf{q}_{k,j}^{\text{dyn}})_h\|^2$ is also strictly positive. The product of three strictly positive terms is strictly positive, which concludes the proof.

$\square$

# D. Appendix. Implementation details

To ensure a fair and robust comparison, we perform an extensive hyperparameter search for our proposed SIT model and all baseline models. For each model, we conduct a comprehensive grid search to identify the optimal set of hyperparameters from the search space defined in Table 4. The combination of parameters yielding the best performance on the validation set was selected for the final evaluation on the test set. To account for stochasticity, all experiments are conducted over five independent runs with different random seeds, and the average performance is reported. For all models and experiments, we maintain a consistent set of general training parameters: the Adam optimizer with a learning rate of $10^{-3}$, a batch size of 64, a dropout rate of 0.1. We train all models for a maximum of 100 epochs, utilizing an early stopping mechanism with a patience of 10 epochs to prevent overfitting.

| *Panel A. General Time Series Forecasting Models* | |
|---|---|
| **Parameter** | **Values** |
| D_MODELS | 32, 64, 128, 256 |
| D_FFS | 32, 64, 128, 256 |
| E_LAYERS_LIST | 1, 2 |
| N_HEADS_LIST | 2, 4, 8 |

| *Panel B. Nonstationary Transformer (NSformer)* | |
|---|---|
| **Parameter** | **Values** |
| D_MODELS | 32, 64, 128, 256 |
| D_FFS | 32, 64, 128, 256 |
| E_LAYERS_LIST | 1, 2 |
| N_HEADS_LIST | 2, 4, 8 |
| P_HIDDEN | 64, 128, 256 |
| P_LAYER | 1,2 |

| *Panel C. TimesNet* | |
|---|---|
| **Parameter** | **Values** |
| D_MODELS | 32, 64, 128, 256 |
| D_FFS | 32, 64, 128, 256 |
| E_LAYERS_LIST | 1, 2 |
| N_HEADS_LIST | 2, 4, 8 |
| TOP_K | 3, 5, 7 |

| *Panel D. RFormer* | |
|---|---|
| **Parameter** | **Values** |
| Embedding_Dim | 8, 16, 32 |
| E_LAYERS_LIST | 1, 2 |
| N_HEADS_LIST | 2, 4, 8 |
| Sig_Level | 2, 3 |

| *Panel E. SIT (Ours)* | |
|---|---|
| **Parameter** | **Values** |
| D_MODELS | 8, 16, 32, 64 |
| D_FFS | 8, 16, 32, 64 |
| E_LAYERS_LIST | 1, 2 |
| N_HEADS_LIST | 2, 4, 8 |
| Sig_Level | 2 |
| HIDDEN_C | 8, 16, 32 |

*Table 4.* The hyperparameter search space for the models used in this study. Each panel shows the parameters and their range of values assigned to a specific model or model group.

# E. Appendix. Why we choose CVaR?

## 1. Model and Definitions

Let $\mathcal{S} = \{1, \ldots, N\}$ be a finite state space for an integer $N \geq 2$. Let $\mathfrak{P}$ be a probability measure on $\mathcal{S}$ assigning a probability $p_s = \mathfrak{P}(\{s\}) > 0$ to each state $s \in \mathcal{S}$, with $\sum_{s=1}^{N} p_s = 1$. We designate state $s = 1$ as the unique **crash state**, with probability $p_1 = q \in (0, 1)$.

We consider two portfolios, a primary portfolio (PF) and a hedged portfolio (HF), with associated losses given by the random variables $X$ and $Y$, respectively. We denote their specific loss values in state $s$ by $X_s$ and $Y_s$. We impose two structural assumptions on these portfolios:

1. **Crash State Exceptionalism:** The loss of the PF portfolio in the crash state is strictly greater than its loss in any non-crash state. That is, $X_1 > X_s$ for all $s \in \{2, \ldots, N\}$.

2. **Strict State-wise Dominance:** The HF portfolio is strictly less risky than the PF portfolio in every state. That is, $Y_s < X_s$ for all $s \in \mathcal{S}$.

For a loss variable $Z$ and a **confidence level** $p \in (0, 1)$, the **Value-at-Risk** is the $p$–quantile

$$\mathrm{VaR}_p(Z) = \inf\{z \in \mathbb{R} \mid \mathfrak{P}(Z \leq z) \geq p\}. \tag{43}$$

The **Conditional Value-at-Risk** (CVaR), also known as Expected Shortfall, at level $\alpha \in (0, 1)$ averages the upper tail of mass $1 - \alpha$:

$$\mathrm{CVaR}_\alpha(Z) = \frac{1}{1 - \alpha} \int_\alpha^1 \mathrm{VaR}_p(Z)\, dp = \min_{\nu \in \mathbb{R}} \left\{ \nu + \frac{1}{1 - \alpha} \mathbb{E}\big[(Z - \nu)^+\big] \right\}. \tag{44}$$

We define the **risk gap** between the two portfolios at level $\alpha$ as

$$\Delta_\alpha := \mathrm{CVaR}_\alpha(X) - \mathrm{CVaR}_\alpha(Y). \tag{45}$$

**Proposition E.1** (HF dominates PF in CVaR). *Let $\alpha \in (0, 1)$ satisfy $1 - \alpha < q$ (equivalently, $\alpha > 1 - q$). For any portfolios PF and HF satisfying the assumptions above, the risk gap is strictly positive and bounded below by the minimum performance gap:*

$$\Delta_\alpha \geq L_{\min}, \tag{46}$$

*where the **minimum performance gap** is defined as*

$$L_{\min} := \min_{s \in \mathcal{S}} (X_s - Y_s). \tag{47}$$

*Since $Y_s < X_s$ for all $s$ in the finite set $\mathcal{S}$, it follows that $L_{\min} > 0$, confirming that HF strictly dominates PF in terms of CVaR for this range of $\alpha$.*

*Proof.* We proceed in three steps. First, we compute $\mathrm{CVaR}_\alpha(X)$ under the stated tail condition. Second, we upper-bound $\mathrm{CVaR}_\alpha(Y)$. Finally, we combine these results.

Exact value of $\mathrm{CVaR}_\alpha(X)$ for $\alpha > 1 - q$. Let $F_X(z) = \mathfrak{P}(X \leq z)$ be the cumulative distribution function of $X$. By Crash State Exceptionalism, $X_1$ is the unique maximum of $X$. Hence, for any $z < X_1$,

$$F_X(z) = \mathfrak{P}(X \leq z) \leq \sum_{s=2}^{N} p_s = 1 - q. \tag{48}$$

Therefore, for every $p \in (1 - q, 1]$, the smallest $z$ with $F_X(z) \geq p$ is $z = X_1$, i.e., $\mathrm{VaR}_p(X) = X_1$. If $\alpha > 1 - q$ (equivalently, the tail mass $1 - \alpha < q$), then

$$\mathrm{CVaR}_\alpha(X) = \frac{1}{1 - \alpha} \int_\alpha^1 \mathrm{VaR}_p(X)\, dp = \frac{1}{1 - \alpha} \int_\alpha^1 X_1\, dp = X_1. \tag{49}$$

Upper bound for $\text{CVaR}_\alpha(Y)$. By definition of $L_{\min}$, we have $X_s - Y_s \geq L_{\min}$ for all $s \in \mathcal{S}$, equivalently

$$Y \leq X - L_{\min} \quad \text{(state-wise)}. \tag{50}$$

Two standard properties of CVaR at a fixed level $\alpha$ are:

1. **Monotonicity:** If $Z_1 \leq Z_2$ state-wise, then $\text{CVaR}_\alpha(Z_1) \leq \text{CVaR}_\alpha(Z_2)$.

2. **Translation Equivariance:** For any constant $c \in \mathbb{R}$, $\text{CVaR}_\alpha(Z - c) = \text{CVaR}_\alpha(Z) - c$.

Applying these to $Y \leq X - L_{\min}$ yields

$$\text{CVaR}_\alpha(Y) \leq \text{CVaR}_\alpha(X - L_{\min}) = \text{CVaR}_\alpha(X) - L_{\min} = X_1 - L_{\min}. \tag{51}$$

So, to get the risk gap, we combine the steps mentioned above.

$$\Delta_\alpha = \text{CVaR}_\alpha(X) - \text{CVaR}_\alpha(Y) \geq X_1 - (X_1 - L_{\min}) = L_{\min} > 0. \tag{52}$$

This completes the proof. $\qquad\square$

## F. Details of Predict-then-Optimize Baselines

The deep learning baselines evaluated in our experiments operate under a two-stage predict-then-optimize approach. Unlike SIT, these baselines treat the two tasks as disjoint stages. This section details the mathematical formulation of this process.

**Stage 1: Return Prediction via MSE** In the first stage, a forecasting model $f_\theta$ is trained to minimize the statistical discrepancy between the predicted returns and the ground truth. Let $\mathbf{X}_t$ denote the lookback window of historical asset features at time $t$, and $\mathbf{r}_{t+1} \in \mathbb{R}^d$ denote the realized returns at time $t + 1$. The model parameters $\theta$ are optimized using the Mean Squared Error (MSE) loss function:

$$\mathcal{L}_{\text{MSE}}(\theta) = \frac{1}{T} \sum_{t=1}^{T} \|\mathbf{r}_{t+1} - \hat{\mathbf{r}}_{t+1}\|_2^2 \tag{53}$$

where $\hat{\mathbf{r}}_{t+1} = f_\theta(\mathbf{X}_t)$ is the point forecast of the asset returns. The training process focuses solely on maximizing predictive accuracy (minimizing $L_2$ distance) without considering the downstream portfolio risk metric or the covariance structure between assets.

**Stage 2: Portfolio Optimization via Mean-CVaR** In the second stage, the trained forecasting model is frozen. Its output $\hat{\mathbf{r}}_{t+1}$ is treated as the vector of expected returns to construct the portfolio. To ensure a fair comparison with our proposed method, we employ a CVaR optimization framework. The solver seeks a portfolio weight vector $\mathbf{w}_t$ that minimizes the Conditional Value-at-Risk (CVaR) while satisfying a target return constraint derived from the prediction $\hat{\mathbf{r}}_{t+1}$.

The optimization problem at time $t$ is formulated as follows:

$$\begin{aligned}
\underset{\mathbf{w} \in \Delta^d, \zeta \in \mathbb{R}}{\text{minimize}} \quad & \zeta + \frac{1}{(1-\alpha)S} \sum_{s=1}^{S} \left[ -(\mathbf{w}^\top \mathbf{r}_s) - \zeta \right]^+ \\
\text{subject to} \quad & \mathbf{w}^\top \hat{\mathbf{r}}_{t+1} \geq \mu_{\text{target}}, \\
& \mathbf{w} \in \mathcal{W}
\end{aligned} \tag{54}$$

Here, $\Delta^d$ represents the simplex of valid portfolio weights (e.g., $\sum w_i = 1, w_i \geq 0$ for long-only strategies). The risk term $\text{CVaR}_\alpha$ is approximated using $S$ historical scenarios $\mathbf{r}_s$ sampled from the immediate past, and $\zeta$ represents the Value-at-Risk (VaR) auxiliary variable.

# G. Appendix. Additional Experiments

| *Panel A. Asset 30 Universe (S&P100)* | | | | |
|---|---|---|---|---|
| Model | Sharpe | Sortino | MDD | Wealth |
| CVaR | 0.2883 | 0.3707 | **0.3499** | 1.1915 |
| EW | 0.5268 | 0.6569 | 0.3724 | 1.5648 |
| GMV | 0.1690 | 0.2177 | 0.2853 | 1.0723 |
| HRP | **0.4609** | **0.5711** | 0.3287 | **1.4099** |
| Autoformer | 0.3228 ± 0.0549 | 0.4500 ± 0.0840 | 0.3782 ± 0.0062 | 1.2989 ± 0.1028 |
| DLinear | 0.3929 ± 0.1294 | 0.5399 ± 0.1758 | 0.3863 ± 0.0266 | 1.4235 ± 0.2587 |
| FEDformer | 0.1594 ± 0.1323 | 0.2162 ± 0.1790 | 0.4345 ± 0.0319 | 1.032 ± 0.2090 |
| iTransformer | 0.2948 ± 0.0721 | 0.3853 ± 0.0942 | 0.4169 ± 0.0118 | 1.2447 ± 0.1459 |
| NSformer | 0.2227 ± 0.1535 | 0.3070 ± 0.2126 | 0.4422 ± 0.0535 | 1.1190 ± 0.2650 |
| PatchTST | 0.2189 ± 0.1446 | 0.2916 ± 0.1945 | 0.5003 ± 0.0667 | 1.1238 ± 0.2287 |
| TimesNet | 0.2192 ± 0.1520 | 0.2999 ± 0.2103 | 0.4434 ± 0.0311 | 1.1213 ± 0.2853 |
| RFormer | 0.4631 ± 0.2771 | 0.5854 ± 0.2094 | 0.4561 ± 0.0501 | 1.5566 ± 0.2038 |
| SIT (Ours) | **0.5496 ± 0.0552** | **0.6797 ± 0.0792** | **0.3415 ± 0.0162** | **1.5678 ± 0.0973** |

*Table 5.* Portfolio performance of SIT versus baselines across 30-asset universes. The best, second-best, and third-best results for each metric are highlighted in red, blue, and **bold**, respectively. SIT consistently delivers superior risk-adjusted returns.

| *Panel A. Asset 10 Universe (DOW30)* | | | | |
|---|---|---|---|---|
| Models | Sharpe Ratio (↑) | Sortino Ratio (↑) | Maximum Drawdown (↓) | Final Wealth Factor (↑) |
| CVaR | 0.4584 | 0.5617 | **0.3053** | 1.4341 |
| EW | 0.9123 | **1.1714** | 0.3191 | **2.4551** |
| GMV | **1.0394** | **1.3191** | **0.2467** | 2.3841 |
| HRP | 0.8407 | 1.0332 | 0.3104 | 2.0583 |
| Autoformer | 0.6767 ± 0.3150 | 0.9581 ± 0.4848 | 0.4655 ± 0.0265 | 2.2787 ± 1.3004 |
| DLinear | 0.8223 ± 0.1251 | 0.9789 ± 0.1692 | 0.4240 ± 0.0414 | **2.4523 ± 0.6336** |
| FEDformer | 0.7664 ± 0.0867 | 0.8245 ± 0.1460 | 0.4948 ± 0.0116 | 2.0578 ± 0.7550 |
| iTransformer | **0.9458 ± 0.1279** | 1.1248 ± 0.2274 | 0.4230 ± 0.0532 | 2.4016 ± 1.7240 |
| NSformer | 0.8863 ± 0.2525 | 0.9630 ± 0.4478 | 0.4733 ± 0.0825 | 2.1044 ± 1.1644 |
| PatchTST | 0.7815 ± 0.1745 | 0.9712 ± 0.2462 | 0.4133 ± 0.0224 | 2.1454 ± 0.8895 |
| TimesNet | 0.4249 ± 0.2673 | 0.5876 ± 0.3658 | 0.6326 ± 0.0967 | 1.6655 ± 0.6016 |
| RFormer | 0.8605 ± 0.1936 | 1.1928 ± 0.2708 | 0.3615 ± 0.0407 | 2.1120 ± 0.5219 |
| SIT (Ours) | **1.0312 ± 0.0671** | **1.3798 ± 0.1049** | **0.2766 ± 0.0413** | **2.8674 ± 0.2263** |

| *Panel B. Asset 20 Universe (DOW30)* | | | | |
|---|---|---|---|---|
| Models | Sharpe Ratio (↑) | Sortino Ratio (↑) | Maximum Drawdown (↓) | Final Wealth Factor (↑) |
| CVaR | 0.5453 | 0.6871 | 0.3249 | 1.5166 |
| EW | **0.8603** | **1.0472** | 0.3503 | **2.2293** |
| GMV | **0.8618** | **1.0730** | **0.2853** | 1.9457 |
| HRP | 0.7500 | 0.8917 | **0.3253** | 1.8443 |
| Autoformer | 0.5688 ± 0.2224 | 0.8312 ± 0.3437 | 0.4642 ± 0.0244 | 1.8605 ± 0.9118 |
| DLinear | 0.7969 ± 0.1057 | 0.9339 ± 0.1475 | 0.3276 ± 0.0415 | **2.1966 ± 0.4046** |
| FEDformer | 0.3341 ± 0.5907 | 0.5471 ± 0.8805 | 0.4671 ± 0.0763 | 1.8039 ± 1.1031 |
| iTransformer | 0.4668 ± 0.2290 | 0.6682 ± 0.3666 | 0.6001 ± 0.0208 | 1.8563 ± 0.9329 |
| NSformer | 0.6541 ± 0.4828 | 0.9751 ± 0.7710 | 0.5620 ± 0.0778 | 2.1464 ± 1.0281 |
| PatchTST | 0.6828 ± 0.1866 | 0.9499 ± 0.2417 | 0.5109 ± 0.0395 | 2.0649 ± 0.4307 |
| TimesNet | 0.2381 ± 0.2584 | 0.3428 ± 0.3871 | 0.4919 ± 0.0511 | 1.2356 ± 0.5723 |
| RFormer | 0.7055 ± 0.1568 | 0.8295 ± 0.1663 | 0.4514 ± 0.1046 | 2.0048 ± 0.6198 |
| SIT (Ours) | **0.8861 ± 0.1243** | **1.0949 ± 0.1607** | **0.3151 ± 0.0181** | **2.2039 ± 0.2983** |

*Table 6.* Portfolio performance of SIT versus baselines across 10 and 20-asset universes from DOW30. The best, second-best, and third-best results for each metric are highlighted in red, blue, and **bold**, respectively. SIT consistently delivers superior risk-adjusted returns.

*Panel A. Asset 50 Universe (CSI300)*

| Models | Sharpe Ratio (↑) | Sortino Ratio (↑) | Maximum Drawdown (↓) | Final Wealth Factor (↑) |
|---|---|---|---|---|
| CVaR | 0.8292 | 1.0038 | 0.1220 | 1.0971 |
| EW | **1.1695** | 1.3671 | 0.1263 | 1.1413 |
| GMV | 1.6717 | 2.1255 | 0.0942 | **1.1766** |
| HRP | 1.7428 | 2.0183 | **0.1136** | 1.1825 |
| Autoformer | 0.5834 ± 0.4666 | 0.5923 ± 0.5446 | 0.2441 ± 0.0779 | 0.9079 ± 0.1257 |
| DLinear | 0.5122 ± 0.3699 | 0.6819 ± 0.5769 | 0.2032 ± 0.0664 | 1.1252 ± 0.1615 |
| FEDformer | 0.3267 ± 0.7165 | 0.4185 ± 0.8655 | 0.2822 ± 0.0371 | 1.0383 ± 0.2970 |
| iTransformer | 0.6161 ± 0.1936 | 0.8566 ± 0.2296 | 0.2001 ± 0.0234 | 1.0204 ± 0.1806 |
| NSformer | 0.3010 ± 0.1859 | 0.4132 ± 0.2395 | 0.2889 ± 0.0689 | 1.0775 ± 0.0704 |
| PatchTST | 0.2789 ± 0.1646 | 0.3368 ± 0.2040 | 0.1913 ± 0.0124 | 1.0394 ± 0.0442 |
| TimesNet | 0.8213 ± 0.1636 | 1.0533 ± 0.1929 | 0.2855 ± 0.0954 | 1.1700 ± 0.2386 |
| RFormer | 0.8867 ± 0.2363 | 1.0921 ± 0.2451 | 0.2579 ± 0.0554 | 1.1665 ± 0.1245 |
| SIT (Ours) | **1.9373 ± 0.0091** | **2.3399 ± 0.1711** | **0.0964 ± 0.0046** | **1.2804 ± 0.0105** |

*Panel B. Asset 100 Universe (CSI300)*

| Models | Sharpe Ratio (↑) | Sortino Ratio (↑) | Maximum Drawdown (↓) | Final Wealth Factor (↑) |
|---|---|---|---|---|
| CVaR | **1.5199** | 2.0863 | 0.1155 | 1.2905 |
| EW | 1.1179 | 1.2660 | 0.1302 | 1.1252 |
| GMV | 1.5365 | **2.0612** | 0.1175 | **1.2724** |
| HRP | 1.2424 | 1.6540 | 0.1229 | 1.2097 |
| Autoformer | 0.5681 ± 0.3129 | 0.6014 ± 0.4024 | 0.2529 ± 0.0206 | 0.9725 ± 0.1396 |
| DLinear | 0.7382 ± 0.4826 | 0.8309 ± 0.4303 | 0.2314 ± 0.0778 | 1.0934 ± 0.3219 |
| FEDformer | 0.4269 ± 0.4916 | 0.5356 ± 0.6167 | 0.2831 ± 0.0344 | 1.0846 ± 0.1694 |
| iTransformer | 0.9865 ± 0.2055 | 1.2495 ± 0.2386 | 0.2492 ± 0.0741 | 1.1169 ± 0.3491 |
| NSformer | 0.5470 ± 0.3975 | 0.7586 ± 0.5326 | 0.2175 ± 0.0851 | 1.1560 ± 0.1387 |
| PatchTST | 0.4650 ± 0.1313 | 0.5551 ± 0.1793 | 0.2547 ± 0.5590 | 1.0809 ± 0.1104 |
| TimesNet | 0.7353 ± 0.1357 | 1.0598 ± 0.1992 | 0.2997 ± 0.0940 | 1.1610 ± 0.2039 |
| RFormer | 1.0267 ± 0.2152 | 1.2359 ± 0.2599 | 0.2111 ± 0.0732 | 1.2321 ± 0.1094 |
| SIT (Ours) | **1.8772 ± 0.0918** | **2.3637 ± 0.0936** | **0.1199 ± 0.0048** | **1.2777 ± 0.0214** |

*Table 7.* Portfolio performance of SIT versus baselines across 10 and 20-asset universes from CSI300. The best, second-best, and third-best results for each metric are highlighted in red, blue, and **bold**, respectively. SIT consistently delivers superior risk-adjusted returns.

## H. Appendix. Drawdown with Gamma($\gamma$)

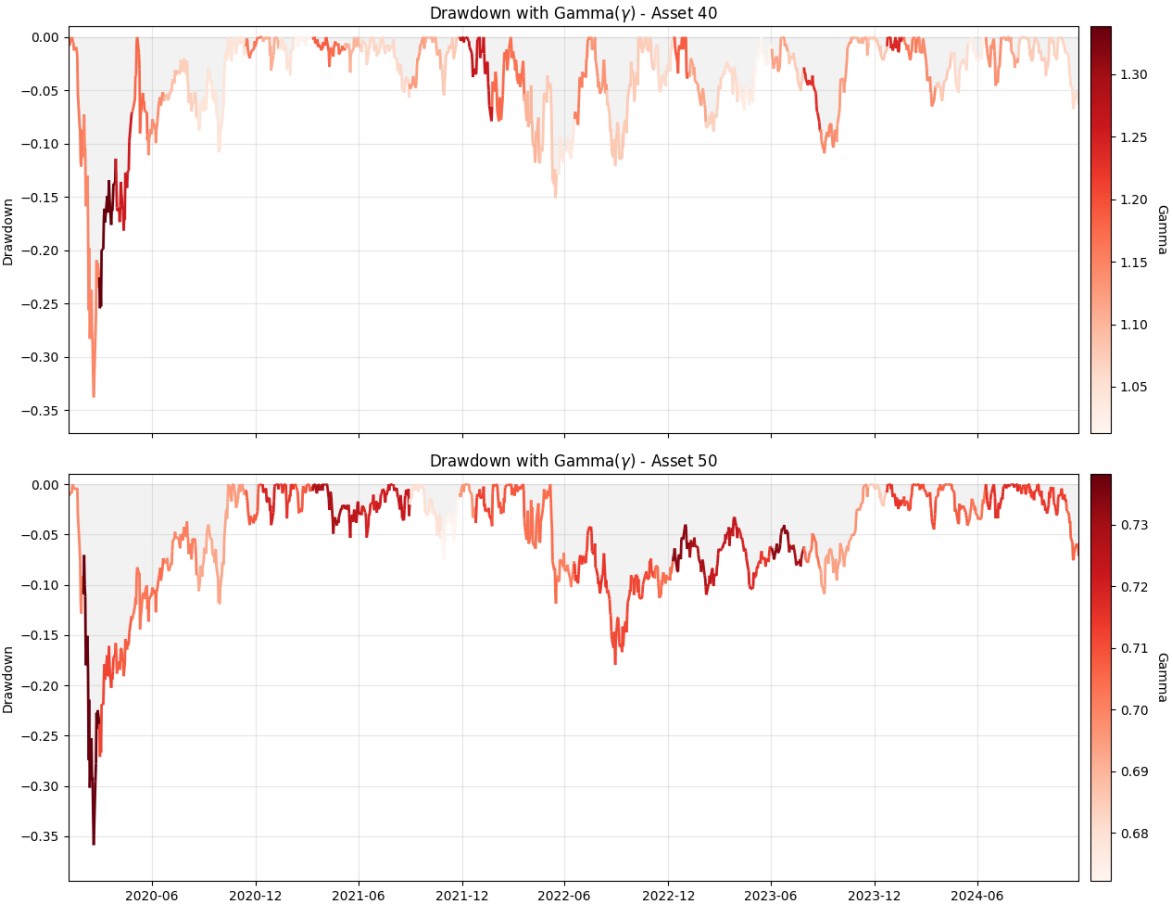

*Figure 7.* Visual analysis of the dynamic gate $\gamma$ relative to portfolio drawdown over the test period (2020-2024). The plots display the drawdown curves for the 40-asset (top) and 50-asset (bottom) universes, where the line color intensity encodes the magnitude of the learnable scalar $\gamma$.

## I. Additional Analysis on Signature Order and Computational Overhead

**Choice of signature truncation order.** We set the signature truncation order to $M = 2$ in the main experiments. This choice is motivated by both representation and efficiency. First-order signature terms summarize only the net increments of each path, whereas second-order terms are the lowest-order terms that encode signed-area geometry and directional lead-lag structure. These second-order interactions are important for the proposed Signature-Informed Transformer (SIT), since the model uses pairwise cross-signature features to bias asset-wise attention.

Table 8 reports a sensitivity analysis over $M \in \{1, 2, 3\}$. Increasing the truncation order from $M = 1$ to $M = 2$ improves the main risk-adjusted performance metrics, including Sharpe, Sortino, and terminal wealth, with moderate additional runtime and memory cost. Although $M = 1$ has the lowest maximum drawdown and computational cost, it does not include second-order signed-area terms and therefore cannot explicitly capture the lead-lag information used by SIT. Increasing the order further to $M = 3$ increases runtime and peak memory without improving the main performance metrics. We therefore use $M = 2$ as the default setting, balancing expressiveness and computational efficiency.

**Computational overhead.** SIT introduces additional computational cost from two sources: (i) per-asset signature embeddings and (ii) pairwise cross-signature features used to construct the signature-informed attention bias. The second component requires modeling asset-pair relations and therefore grows with the number of assets. Table 9 reports the empirical memory usage and runtime as the asset universe increases. The overhead is moderate at the asset scales used in the main experiments, but grows for larger asset universes, as expected from the pairwise relational structure. This cost is

*Table 8.* Sensitivity analysis with respect to the signature truncation order $M$. ↑ indicates that higher is better and ↓ indicates that lower is better. Runtime and peak memory are normalized relative to $M = 1$.

| Signature order | Sharpe ↑ | Sortino ↑ | MDD ↓ | Wealth ↑ | Runtime / epoch ↓ | Peak memory ↓ |
|---|---|---|---|---|---|---|
| $M = 1$ | $0.617 \pm 0.049$ | $0.776 \pm 0.064$ | $\mathbf{0.348 \pm 0.008}$ | $1.699 \pm 0.083$ | $\mathbf{1.00 \pm 0.02}$ | $\mathbf{1.00 \pm 0.01}$ |
| $M = 2$ | $\mathbf{0.672 \pm 0.063}$ | $\mathbf{0.823 \pm 0.079}$ | $0.361 \pm 0.004$ | $\mathbf{1.790 \pm 0.102}$ | $1.18 \pm 0.03$ | $1.21 \pm 0.02$ |
| $M = 3$ | $0.660 \pm 0.072$ | $0.810 \pm 0.089$ | $0.365 \pm 0.009$ | $1.768 \pm 0.118$ | $1.53 \pm 0.04$ | $1.58 \pm 0.03$ |

*Table 9.* Empirical computational overhead of SIT compared with a vanilla Transformer as the number of assets increases. A dash indicates that the corresponding SIT result was not reported under the same experimental setting.

| Assets | SIT Mem. (GiB) | Transformer Mem. (GiB) | SIT Time (s) | Transformer Time (s) |
|---|---|---|---|---|
| 50 | 1.81 | 1.174 | 3.91 | 3.050 |
| 100 | 5.78 | 2.190 | 15.65 | 7.561 |
| 200 | 21.67 | 6.255 | 62.60 | 21.391 |
| 300 | 48.14 | 13.030 | 140.84 | 41.631 |
| 400 | 85.21 | 22.515 | 250.38 | 68.281 |
| 500 | – | 34.710 | – | 101.341 |

the price of explicitly modeling cross-asset lead-lag information through signature-informed attention. In the component ablation, the signature-informed financial bias improves the Sharpe ratio from $0.6047$ without the Financial Bias to $0.7715$ with it, indicating that the additional cost is accompanied by a meaningful performance gain.

## J. Additional Results for the 50-Asset Universe

Table 10 reports the full out-of-sample portfolio evaluation results after CVaR portfolio optimization on the 50-asset S&P100 universe. We report the Sharpe ratio, Sortino ratio, maximum drawdown (MDD), and terminal wealth. Higher values are better for Sharpe, Sortino, and Wealth, while lower values are better for MDD.

*Table 10.* Out-of-sample portfolio evaluation results after CVaR portfolio optimization on the 50-asset S&P100 universe. The best performance for each metric is highlighted in bold.

| Model (Asset 50) | Sharpe ↑ | Sortino ↑ | MDD ↓ | Wealth ↑ |
|---|---|---|---|---|
| Autoformer-DFL | $0.487 \pm 0.067$ | $0.676 \pm 0.095$ | $0.402 \pm 0.061$ | $1.598 \pm 0.173$ |
| DLinear-DFL | $0.572 \pm 0.009$ | $0.790 \pm 0.019$ | $0.351 \pm 0.012$ | $1.562 \pm 0.070$ |
| FEDformer-DFL | $0.637 \pm 0.067$ | $0.876 \pm 0.101$ | $0.364 \pm 0.028$ | $1.761 \pm 0.128$ |
| iTransformer-DFL | $0.717 \pm 0.012$ | $0.933 \pm 0.040$ | $0.339 \pm 0.016$ | $1.905 \pm 0.094$ |
| NSformer-DFL | $0.492 \pm 0.031$ | $0.676 \pm 0.047$ | $0.470 \pm 0.041$ | $1.742 \pm 0.086$ |
| PatchTST-DFL | $0.657 \pm 0.023$ | $0.904 \pm 0.053$ | $0.347 \pm 0.023$ | $1.815 \pm 0.104$ |
| TimesNet-DFL | $0.502 \pm 0.113$ | $0.687 \pm 0.151$ | $0.448 \pm 0.066$ | $1.625 \pm 0.244$ |
| RFormer-DFL | $0.657 \pm 0.229$ | $0.905 \pm 0.255$ | $0.358 \pm 0.074$ | $1.884 \pm 0.246$ |
| **SIT (Ours)** | $\mathbf{0.771 \pm 0.062}$ | $\mathbf{0.974 \pm 0.099}$ | $\mathbf{0.327 \pm 0.009}$ | $\mathbf{1.921 \pm 0.179}$ |

### J.1. Sensitivity to the CVaR Confidence Level

We conduct an additional sensitivity analysis to examine how the confidence level of the portfolio-level $\text{CVaR}_\alpha$ objective affects the behavior of SIT. The confidence level $\alpha$ controls the severity of the tail events emphasized during training: larger values of $\alpha$ place more weight on extreme downside outcomes and therefore encourage more conservative allocations, whereas smaller values allow the model to pursue more aggressive return profiles. We sweep $\alpha \in \{0.8, 0.9, 0.95, 0.99\}$ while keeping all other experimental settings fixed, including the long-only fully-invested constraint, monthly rebalancing, and the same evaluation protocol used in the main experiments. Gross performance is computed before transaction costs, while net performance is computed under a proportional transaction cost of 10 bps.

Table 11 shows a consistent risk–return trade-off across the tested confidence levels. Increasing $\alpha$ makes the learned policy more conservative, as reflected by the monotonic reduction in maximum drawdown from $\alpha = 0.8$ to $\alpha = 0.99$ in both gross and net evaluations. In contrast, smaller values of $\alpha$ allow the model to pursue higher terminal wealth, but at the cost of larger drawdowns. For example, $\alpha = 0.8$ achieves the highest gross wealth, while $\alpha = 0.99$ achieves the lowest gross and net MDD. The intermediate setting $\alpha = 0.95$ provides the most favorable balance between risk control and return

*Table 11.* Sensitivity of SIT to the CVaR confidence level $\alpha$ on the 40-asset S&P100 universe. Values are reported as mean $\pm$ standard deviation. $\uparrow$ indicates that higher is better, and $\downarrow$ indicates that lower is better. Bold values denote the best result in each column. Net results are computed with a transaction cost of 10 bps.

| $\alpha$ | Gross Sharpe ($\uparrow$) | Gross Sortino ($\uparrow$) | Gross MDD ($\downarrow$) | Gross Wealth ($\uparrow$) | Net Sharpe ($\uparrow$) | Net Sortino ($\uparrow$) | Net MDD ($\downarrow$) | Net Wealth ($\uparrow$) |
|---|---|---|---|---|---|---|---|---|
| 0.80 | $0.638 \pm 0.073$ | $0.781 \pm 0.094$ | $0.398 \pm 0.015$ | $\mathbf{1.826} \pm 0.129$ | $0.586 \pm 0.069$ | $0.716 \pm 0.088$ | $0.409 \pm 0.017$ | $1.714 \pm 0.118$ |
| 0.90 | $0.661 \pm 0.067$ | $0.809 \pm 0.086$ | $0.378 \pm 0.011$ | $1.811 \pm 0.114$ | $0.620 \pm 0.063$ | $0.757 \pm 0.081$ | $0.387 \pm 0.012$ | $\mathbf{1.738} \pm 0.107$ |
| 0.95 | $\mathbf{0.672} \pm 0.063$ | $\mathbf{0.823} \pm 0.079$ | $0.361 \pm 0.004$ | $1.790 \pm 0.102$ | $\mathbf{0.638} \pm 0.059$ | $\mathbf{0.781} \pm 0.075$ | $0.369 \pm 0.006$ | $1.733 \pm 0.098$ |
| 0.99 | $0.649 \pm 0.061$ | $0.804 \pm 0.077$ | $\mathbf{0.333} \pm 0.006$ | $1.711 \pm 0.097$ | $0.621 \pm 0.058$ | $0.773 \pm 0.073$ | $\mathbf{0.340} \pm 0.007$ | $1.663 \pm 0.093$ |

generation. It achieves the best gross Sharpe and Sortino ratios, as well as the best net Sharpe and Sortino ratios after transaction costs. This indicates that the main performance of SIT is not driven by a fragile choice of the CVaR confidence level. Rather, the model exhibits a stable and interpretable dependence on $\alpha$: lower values favor wealth accumulation, higher values emphasize drawdown control, and $\alpha = 0.95$ delivers the strongest risk-adjusted performance. We also repeated the same sweep on the 50-asset universe. The results exhibit the same qualitative pattern: larger $\alpha$ values reduce drawdowns, smaller $\alpha$ values increase terminal wealth, and the intermediate setting remains the most balanced in terms of risk-adjusted performance. We omit the full 50-asset table for conciseness.

