# OpenReview forum: "Signature-Informed Transformer for Asset Allocation"
_ICML.cc/2026/Conference — ICML 2026 regular_

### Official Review · Reviewer_z9T7 · 2026-02-22

**Soundness:** 3
**Presentation:** 2
**Significance:** 2
**Originality:** 3
**Overall Recommendation:** 4
**Confidence:** 2

**Summary:**

In this paper, authors propose a new framework called SIT to improve the performance of asset allocations. They propose that the current strategy of separating the solution into a two-stage modelling, where the first stage predicts the value by MSE and the make decisition based on the value-predictor, introduces a mismatch between the objectives of the two modelling stages, as the precision of value-prediction can not directly improve the return. Besides directly considering an end-to-end supervised by CVaR, authors also propose a novel SIT block, which embeds the first and second order signature information into the inputs to the self-attention layer.

**Compliance With Llm Reviewing Policy:**

Affirmed.

**Key Questions For Authors:**

1. Despite the metrics used to evaluate the different allocations, I wonder how the authors expect their model to perform in real-world investment scenarios. For instance, have the authors backtested or deployed their model as a practical trading strategy in live markets? Furthermore, have they considered evaluating their approach on international exchanges beyond the Nasdaq or the S&P 500, such as the Chinese stock market (e.g., SSE/SZSE) or the South Korean market (KOSPI)?

2. In addition, while I'm not familiar with quantitative finance, I feel that only extracting information from historical stock prices might not be sufficient. What about adding other features like the trading volume, the basic information of the company, and maybe other information?

3. While I agree that the combination of two stages of the predict-then-optimize procedure may introduce a mismatch. I'm not sure whether the prediction model can be considered as an encoding layer, i.e., treating the hidden-layer output as a part of the input of the SIT block. Have authors tried on this?

**Limitations:**

From my perspective, the current modelling only extract feactures from the historical prices of different assets, and does not consider some other information like the trading volume or the company fundamentals. In addition, it directly treats each time point as different sample in a batch and drops the potential to introduce a self-attention structure to extract the temporal correlations (which I feel is the cause of the claimed complexity). I feel these points present limitations while not discussed.

**Strengths And Weaknesses:**

**Strengths:**
1. Introduction of the signature is an interesting and novel point for embedding the potential lead-lag behaviors between the different assets.
2. Compared to the baselines presented, the SIT supervised by CVaR achieves the best performance for almost all different evaluation metrics.

**Weaknesses:**
1. Both theorems seem trivial and unrelated to the proposed method. The logical connections are unclear, and it is difficult to see any essential implications. Specifically, the first proposition is established on an extremely impractical strict lead-lag structure assumption. I do not understand how such a strong assumption, where one asset can only change after another, and reaches the same value after a two-stage change, can be verified. Based on this strong assumption, the derivation of Proposition 2.1 seems extremely trivial. In addition, I do not understand why the authors chose to introduce such a proposition. In addition, this value is not directly the second-order signature (only the difference), and it seems that whether it is positive or negative does not affect the model architecture. For Theorem 2.2, the conclusion is also a direct result of calculus. The sign directly arises from the model design instead of any mysterious training result. The analysis in Section 3.3 is also confusing: if there is no regularization to the bias term $b$, why should I feel surprised to see that the correlation between $a+b$ and $b$ is positive?

2. The claimed complexity improvements might be incorrect. First of all, why does $d_{model}$ not appear in the complexity? In addition, besides the additive bias term, the architecture of the SIT block is identical to that of a standard MHA layer. How can the authors claim an improved complexity? What is the source of this improvement? From my perspective, I do not feel that the squared batch size should appear in the final complexity.

3. While I'm not familiar with quantitative finance, the current modelling only extracts features from the historical prices, which may not be sufficient.

As I stated, my expertise lies in the theory, and I'm not familiar with the applications and domain knowledge of finance. I have set my confidence to 2 correspondingly.

---

> ### Author Rebuttal · Authors · 2026-03-25
>
> We thank the reviewer for the careful reading. We agree with two core points upfront:
> - (i) Proposition 2.1/2.2 were framed too strongly in the current draft, and
> - (ii) the complexity statement should be corrected.
>
> **Answer 1: On Proposition** We agree that Proposition 2.1 is a stylized construction, not a realistic market assumption, and we will revise the wording accordingly. Its role is only to motivate why the level-2 signature term is the lowest-order path feature that captures directional temporal asymmetry. We also agree that ($\mathcal A(\mathbf X)$) is not the full second-order signature tensor, but its antisymmetric cross-term. We will state this more precisely. The model does *not* assume strict lead-lag in real data, and it does *not* use the proposition’s sign as a hard-coded rule. The full cross-signature vector, including sign and magnitude, is passed to ($\mathrm{MLP}_\beta$).
>
> We also agree that Proposition 2.2 is a local calculus fact. We did not intend it as proof that signatures are uniquely superior. It only formalizes how the additive bias can affect attention locally. We will tone this down. Likewise, we agree that the correlation analysis in Section 3.3 is *supportive*, not causal. We will revise that section and treat the plot only as evidence that the bias term is not ignored.
>
> The stronger evidence is empirical. Removing the financial bias lowers Sharpe from 0.672 to 0.606 on 40 assets and from 0.772 to 0.607 on 50 assets. More importantly, replacing the signature features with shuffled signatures or random bias features hurts further (see our response to Reviewer DQ8f, Answer 5). On 40 assets, Sharpe drops to 0.560 and 0.547 also, on 50 assets, to 0.572 and 0.554. This is the main reason we believe the gain comes from meaningful signature geometry rather than from the mere presence of an auxiliary additive bias. We also evaluated the truncation order (M) and found that ($M$=2) gives the best trade-off; see our response to Reviewer 6ejy, Answer 1.
>
> **Answer 2: On the complexity** The reviewer is right that the current complexity statement is incomplete. The asymptotic improvement does not come from the additive bias itself. It comes from using factorized attention over time and assets instead of flattened attention over (H $\times$ d) tokens. Therefore, ($d_{\text{model}}$) should appear in the complexity, and the signature bias adds an extra quadratic cross-asset overhead. We will correct this explicitly in the revision.
>
> More precisely, the flattened attention cost is ($O(H^2 d^2 d_{\text{model}})$), while the factorized Transformer part scales as ($O(dH^2 d_{\text{model}} + Hd^2 d_{\text{model}})$), plus the signature overhead from pairwise cross-signatures computation. Empirically, the main bottleneck is indeed the pairwise cross-signature. Please see our response to Reviewer DQ8f, Answer 2 and Reviewer 6ejy, Answer 2 for runtime/memory profiling up to 400 assets.
>
> **Answer 3:  On real-world investment scenarios** Our current evidence is based on out-of-sample backtests, not live deployment. (But we've already got a live demo version ready, and it'll be going public soon!) In addition to S&P100 and DOW30, the appendix already includes CSI300 experiments with 50 and 100 assets. SIT achieves Sharpe 1.937 on 50 assets and 1.877 on 100 assets, with low drawdowns, which suggests that the method is not tied to one market structure. For practical plausibility, we also report transaction-cost sensitivity and turnover-related diagnostics. Please see our response to Reviewer DQ8f, Answer 3: average turnover is 34.57% (40 assets) and 35.52%(50 assets), and SIT remains robust under 0/5/10 bps one-way trading costs.
>
> **Answer 4:  On using only historical prices**  Strictly speaking, the current model is not purely price-only because it already includes calendar features (\mathbf v_t), but it does not yet use volume, fundamentals, or other side information. This was a deliberate choice to isolate the value of path geometry and cross-asset relational structure. The architecture can naturally absorb richer inputs: time-varying features such as volume or realized volatility can be added at the slice level, and static company descriptors such as fundamentals or sector labels can be integrated through additional asset-level embeddings.
>
> **Answer  5. On using a forecasting model as an encoder for SIT.** This is a very good suggestion. That said, SIT is fully compatible with it. Our current DFL results already show that generic forecasting backbones improve when placed inside an end-to-end CVaR allocation pipeline (see our response to Reviewer DQ8f, Answer 1 and Reviewer 6ejy, Answer 3). In that sense, our claim is not that forecasting encoders are useless, but that *MSE-trained forecasting alone is insufficient for allocation*, and that it can be strengthened by decision-focused training plus explicit cross-asset inductive bias.

---

> > ### Author Rebuttal · Reviewer_z9T7 · 2026-04-01
> >
> > I thank the authors for their detailed rebuttal. I believe that most of my concerns have been adequately addressed. While I still feel that the discussion regarding the theorems and complexity has some weaknesses, I do not view these issues as central to the main contribution of the paper, and the authors have indicated that they will revise this part accordingly. Therefore, I am satisfied with the current rebuttal and will raise my score to 4.
> >
> > I still have a few open questions, though they will not affect my evaluation of this work. In particular, I wonder whether the squared dependence on the number of stocks may limit the practical scalability of the model. I suspect this may also explain why the experiments focus on indices such as the Nasdaq 100 or the Dow 40. If so, this could mean that the model is mainly suitable for tracking large-cap stocks, such as the Magnificent 7. However, at least over the past half year, that does not seem to have been the best choice. I am not asking for additional experiments here; this is simply out of curiosity.
> >
> > In addition, I am also curious about several details of the backtesting setup. For example, is the model operated at a daily frequency? More specifically, is it run once before the market opens each day to decide whether the current portfolio should be adjusted? If so, what price point within the day is used to simulate trade execution in the experiments?
> >
> > More broadly, I would also be interested in how this method compares with other quantitative finance approaches. Although I am not very familiar with the mainstream methods in this area, I am curious about the relative strengths and weaknesses of this approach, and whether it could be combined with other existing methods.
> >
> > Finally, given the current popularity of trading agents, I am also curious how a locally deployed quantitative model of this kind could be used together with LLM-based tools in practice.

---

> > > ### Author Response · Authors · 2026-04-01
> > >
> > > **We sincerely thank the reviewer for acknowledging our response and for increasing the score!**
> > >
> > > We also appreciate the additional comments and address them as follows.
> > >
> > > **1. Scalability with respect to the number of assets**
> > > In practice, institutional investors typically pre-screen their investment universe based on various criteria such as regulatory requirements, sector allocations, and internal guidelines. The resulting asset pool usually consists of fewer than 200 securities, and often significantly fewer. At the individual investor level, portfolios are constructed with an even more concentrated set of assets. Therefore, we believe that our framework is unlikely to face scalability limitations in real-world asset management settings.
> > >
> > > **2. Application to index tracking**
> > > For thematic ETFs such as ARKK, LIT, and BOTZ, exploiting the lead lag effect can be particularly effective. If the returns of such ETFs can be reasonably replicated using a smaller subset of assets, individual investors can benefit from substantial fee reductions. However, when tracking ETFs composed primarily of already leading stocks, such as those dominated by the M7, there is a possibility that lagging stocks may be selected instead. In contrast, for thematic ETFs, it is possible to appropriately balance allocations between leading and lagging stocks during asset selection. Under such conditions, we expect our methodology to perform effectively.
> > >
> > > **3. Backtesting setup**
> > > Daily rebalancing leads to excessive transaction costs, so we adopted a 20 day rebalancing cycle to better reflect institutional practices. As noted in Reviewer DQ8f’s Answer 3, the turnover of our model is stable enough to withstand transaction costs of 10 basis points. In practice, order execution algorithms would also be required to minimize market impact. However, since our primary focus is portfolio optimization, large scale order execution falls outside the main scope of this work and should be considered as a post hoc component when applying the model in real trading.
> > >
> > > **4. Comparison with traditional methods such as MVO**
> > > Interestingly, many institutions still prefer well established traditional methods, and approaches such as equal weighting and mean variance optimization remain widely used. However, MVO is highly sensitive to small errors in expected return estimates, which can result in extreme portfolio weights, and in practice these are often manually adjusted [1,2]. One may then consider using AI to directly predict returns. As shown in Tables 1, 5, 6, and 7, the predict then optimize approach using well known time series models exhibits very large standard deviations. This indicates that portfolio weights can fluctuate significantly depending on the random seed. In contrast, when the model is designed to directly learn the constraints of the optimization process, the standard deviation is substantially reduced, leading to stable and consistent weights across repeated runs. Nevertheless, traditional methods such as MVO and CVaR provide closed form solutions and therefore guarantee identical results, whereas deep learning models inherently involve some degree of uncertainty. Despite this limitation, our model demonstrates highly promising performance compared to traditional approaches. We expect that combining it with order execution algorithms designed to mitigate market impact in large scale trading could further enhance its effectiveness.
> > >
> > > **5. Look ahead bias in agent based research**
> > > Many existing agent based studies overlook the time lag between disclosure dates and the actual incorporation of information into the market, leading to unrealistic Sharpe ratios affected by look ahead bias. As a result, generating meaningful alpha through daily trading with such agents is quite challenging in practice [3,4]. However, since institutional investors do not typically engage in daily trading, this bias can be minimized. By incorporating unstructured data alongside structured data, the lead lag effect, which is central to our framework, can be captured more accurately.
> > >
> > > Once again, we sincerely thank the reviewer for their careful evaluation and valuable suggestions.
> > >
> > > [1] Michaud, R. O. (1989). The Markowitz optimization enigma: Is ‘optimized’optimal?. Financial analysts journal, 45(1), 31-42.
> > >
> > > [2] Chung, M., Lee, Y., Kim, J. H., Kim, W. C., & Fabozzi, F. J. (2022). The effects of errors in means, variances, and correlations on the mean-variance framework. Quantitative Finance, 22(10), 1893-1903.
> > >
> > > [3] https://nof1.ai/leaderboard
> > >
> > > [4] https://llm-stats.com/arenas/trading-arena

---

### Official Review · Reviewer_cbzZ · 2026-03-02

**Soundness:** 1
**Presentation:** 3
**Significance:** 3
**Originality:** 3
**Overall Recommendation:** 3
**Confidence:** 5

**Summary:**

This paper proposes an end-to-end portfolio allocation framework called Signature-Informed Transformer (SIT), aiming to address target mismatch in the traditional predict-then-optimize paradigm. The authors argue that training only on prediction error cannot guarantee robust portfolio performance. To fix this, they build a unified decision-learning model that integrates feature extraction, asset relationship modeling, and portfolio construction in a single framework. The model uses path signatures to capture higher-order geometric features of asset prices, and introduces a signature-enhanced attention mechanism in the Transformer. This explicitly embeds lead–lag structures between assets into the attention calculation. At the same time, the training objective directly minimizes portfolio CVaR, aligning the learning target with risk-control goals. Empirical results show that the proposed method outperforms traditional strategies and prediction-based deep learning models on several stock universes, achieving better risk-adjusted returns.

**Compliance With Llm Reviewing Policy:**

Affirmed.

**Final Justification:**

Overall, I feel the contributions of this paper somewhat scattered and not well integrated. For example, the use of signatures and the introduction of the CVaR loss are not organically connected, and the overall method feels more like a patchwork of components. Concretely, the connection between the proposed method and the underlying theory is relatively weak; the analysis of how signatures contribute to the attention mechanism is insufficient; and the reliability of directly optimizing portfolio positions, compared to traditional portfolio optimization, requires further validation (although the authors provide a simple experiment during the rebuttal, I am unable to make a confident judgment given the limited time). Thus, I will maintain my current score.

**Key Questions For Authors:**

see weaknesses

**Limitations:**

yes

**Strengths And Weaknesses:**

Strengths:
1. The paper clearly points out the issues of “target mismatch” and error amplification in the predict-then-optimize paradigm. Starting from the real objective of portfolio allocation, it emphasizes risk-adjusted returns rather than prediction accuracy.
2. By introducing path signatures to describe the higher-order geometric features of asset price trajectories and injecting a signature-based bias into the attention mechanism, the model embeds financial lead–lag structures into the Transformer architecture in an interpretable way.
3. Instead of relying on prediction loss, the model directly minimizes portfolio-level CVaR, so the training objective is aligned with the final decision objective. This makes the approach logically consistent and theoretically sound for portfolio allocation.

Weakness:
1. The paper highlights its contribution by criticizing the traditional prediction-then-optimization paradigm. However, in deep portfolio management, many existing methods already output portfolio weights directly and optimize risk-adjusted returns end-to-end (e.g., RL-based allocation, other differentiable portfolio optimization). These approaches also avoid the mismatch between prediction loss and portfolio objectives and have been well studied. Therefore, comparing mainly with two-stage prediction models makes it hard to show the unique advantage of the proposed method in addressing the prediction–optimization mismatch. A more thorough comparison with existing end-to-end direct-allocation models would better support the paper’s novelty.
2. The paper overstates the ability of end-to-end optimization to solve the “objective mismatch” problem, while overlooking its rigidity in real-world decision-making. Compared with traditional two-stage “prediction–optimization” approaches, the SIT model hard-codes a specific risk preference (e.g., CVaR loss) into the model parameters, making it difficult to adjust to investors’ changing risk aversion without retraining. It would also be helpful to discuss how the choice of the CVaR confidence level $\alpha$ under different application scenarios affects the empirical results.
3. Directly outputting portfolio weights through a black-box model sacrifices interpretability, which is important in financial decision-making. It also makes it harder to handle practical hard constraints—such as industry neutrality, turnover limits, or single-stock weight caps—as rigorously and efficiently as traditional optimizers. This weakens the model’s practicality in real investment settings. The authors are therefore encouraged to further elaborate, within the current framework, how such risk and constraint controls could be implemented without reverting to a traditional optimization.
4. The paper attempts to use Proposition 2.2 to argue that signature features drive the attention mechanism, but this proposition is logically very weak. In essence, it only reflects the structural monotonicity of the Softmax function when an additive bias term is introduced, rather than any specific exploitation of financial lead–lag relationships. As long as the model architecture includes an additive bias, this derivative property is almost a trivial consequence and does not demonstrate that Path Signatures are superior to other features in guiding the learning of asset relationships.
5. The paper tries to justify the effectiveness of its Signature-Informed Attention by showing a positive correlation between the path signature features $s$ and the final attention scores $a$. However, in the SIT architecture, the attention score is a mixture jointly determined by a standard dot-product term based on raw features and a bias term $B$ generated from signature features. Since the standard dot-product term itself may already capture market signals that are highly correlated with path features, the correlation between $s$ and $a$ alone cannot prove that the bias term $B$ plays the dominant role. It is recommended to add ablation experiments to show how the attention distribution changes substantially when the $B$ term is removed.

---

> ### Author Rebuttal · Authors · 2026-03-25
>
> We thank the reviewer for the constructive comments.
>
> **Answer W1:** We agree that direct allocation and decision-focused portfolio learning are not new in themselves. Our contribution is more precise: SIT combines a portfolio-level CVaR objective with a signature-informed cross-asset attention mechanism, allowing geometric priors to be incorporated directly into the allocation model.
>
> The two-stage baselines are included primarily to illustrate the prediction–optimization mismatch. Independently of this motivation, our architectural contribution is validated by two controlled comparisons already presented in the paper. First, Figure 4 demonstrates that, even with the same backbone, decision-focused training consistently outperforms MSE-based training in terms of allocation quality. Additional evidence is provided in Reviewer DQ8f, Answer 1. Second, Table 2 shows that, under an identical portfolio objective, the signature-informed architecture further improves performance over a matched attention backbone that does not incorporate the same financial inductive bias. Additional supporting results are provided in Reviewer DQ8f, Answer 5.
>
> Importantly, the gains of SIT are not explained simply by the inclusion of signature features. RFormer also uses signatures, but SIT outperforms RFormer by using signatures for decision-oriented allocation rather than just forecasting.
>
> **Answer W2:**  In response to the query regarding the CVaR confidence level, we conducted a sensitivity analysis using $\alpha \in \{0.8, 0.9, 0.95, 0.99\}$. The findings demonstrate a consistent risk-return trade-off across both the 40- and 50-asset universes. Specifically, higher values of $\alpha$ lead to a more conservative strategy with reduced drawdowns, whereas lower values of $\alpha$ allow for a more aggressive profile with higher cumulative returns. Note that the *Net* results reflect an applied transaction cost of 10 bps. We confirmed identical patterns for the 50-asset universe. however, due to space constraints, these results are omitted here but can be provided upon request
>
> | (\alpha) | Gross Sharpe ($\uparrow$) | Gross Sortino ($\uparrow$) | Gross MDD ($\downarrow$) | Gross Wealth ($\uparrow$) | Net Sharpe ($\uparrow$) | Net Sortino ($\uparrow$) | Net MDD ($\downarrow$) | Net Wealth ($\uparrow$) |
> | -------- | ----------------------: | -----------------------: | ---------------------: | ----------------------: | --------------------: | ---------------------: | -------------------: | --------------------: |
> | 0.80     |           0.638 ± 0.073 |            0.781 ± 0.094 |          0.398 ± 0.015 |       **1.826 ± 0.129** |         0.586 ± 0.069 |          0.716 ± 0.088 |        0.409 ± 0.017 |         1.714 ± 0.118 |
> | 0.90     |           0.661 ± 0.067 |            0.809 ± 0.086 |          0.378 ± 0.011 |           1.811 ± 0.114 |         0.620 ± 0.063 |          0.757 ± 0.081 |        0.387 ± 0.012 |     **1.738 ± 0.107** |
> | **0.95** |       **0.672 ± 0.063** |        **0.823 ± 0.079** |          0.361 ± 0.004 |           1.790 ± 0.102 |     **0.638 ± 0.059** |      **0.781 ± 0.075** |        0.369 ± 0.006 |         1.733 ± 0.098 |
> | 0.99     |           0.649 ± 0.061 |            0.804 ± 0.077 |      **0.333 ± 0.006** |           1.711 ± 0.097 |         0.621 ± 0.058 |          0.773 ± 0.073 |    **0.340 ± 0.007** |         1.663 ± 0.093 |
>
> **Answer W3:**  We agree on the importance of interpretability and constraint handling. Currently, long-only and fully invested constraints are directly enforced. Other practical constraints (e.g., single-stock caps, turnover control, and sector neutrality) can be integrated via differentiable projections or penalties without reverting to a traditional optimizer. While our model lacks the full transparency of a convex optimizer, the signature bias and gate offer partial interpretability into cross-asset relations. We will clarify this scope in the revision.
>
> **Answer W4:**  We agree that Proposition 2.2 is logically modest and does not prove signature superiority. it merely formalizes the local mechanism of the additive bias. Similarly, Proposition 2.1 is intended as a stylized illustration of second-order terms. In the revision, we will reduce the rhetorical weight of these propositions, move part of the theory to the appendix, and clarify that our model's primary validation stems from empirical results and ablations.
>
> **Answer W5**: We agree that the correlation between signature strength and final attention mass is supportive evidence, not causal proof. Accordingly, we have included new experimental results, such as the 'Answer 5. SIT component' mentioned in our response to Reviewer DQ8f. These variants consistently underperform both the full SIT model and the no-bias baseline. This demonstrates that the performance gains stem from the meaningful geometry of the signature, rather than simply from increased model capacity or the mere presence of an auxiliary bias.

---

> > ### Author Rebuttal · Reviewer_cbzZ · 2026-04-01
> >
> > Thank you for the detailed response, which partially addresses my concerns. However, overall, the paper still lacks a comparison with other end-to-end direct-allocation models. In addition, although the authors claim that “Other practical constraints (e.g., single-stock caps, turnover control, and sector neutrality) can be integrated via differentiable projections or penalties without reverting to a traditional optimizer,” this remains to be validated. I understand that quantitative finance research does not necessarily need to include detailed risk optimization or industry-grade portfolio analysis. But given that one of the paper’s claimed contributions is direct allocation, the above analyses are necessary. Otherwise, I will keep my current score.

---

> > > ### Author Response · Authors · 2026-04-02
> > >
> > > We sincerely appreciate your constructive feedback. We apologize for any lack of clarity in our initial manuscript and we will ensure that the related work and appendix sections are thoroughly updated to reflect our design philosophy explicitly. Due to length limitations, not all experimental results are shown, though similar outcomes were observed overall. References are also limited to titles only due to space constraints.
> > >
> > > ---
> > > **Regarding the Comparison with End to End Direct Allocation Models**  We sincerely apologize for the unclarity. We will revise the appendix to include the relevant results and clearly explain our philosophy.True end-to-end direct-allocation models are actually quite rare. Most existing frameworks [1-3] focus on directional trading rather than portfolio optimization. Other models claim an end-to-end approach but optimize the Sharpe Ratio without directly outputting the portfolio weight $w$ [4-5]. While some studies [6, 7] compute weights directly, they prioritize optimization using simple models like DLinear which completely lack financial inductive biases.Furthermore, advanced forecasting models excel on financial datasets but perform poorly under predict-then-optimize frameworks for determining $w$. As shown in our table, applying a genuine end-to-end direct-allocation approach to these models significantly improves the standard deviation and yields highly meaningful results. Our study ultimately emphasizes making optimal decisions for the actual problem over merely competing in forecasting accuracy.
> > >
> > > | Model (Asset 50)      |          Sharpe ↑ |         Sortino ↑ |             MDD ↓ |          Wealth ↑ |
> > > | ---------------- | ----------------: | ----------------: | ----------------: | ----------------: |
> > > | Autoformer-DFL   |     0.487 ± 0.067 |     0.676 ± 0.095 |     0.402 ± 0.061 |     1.598 ± 0.173 |
> > > | DLinear-DFL      |     0.572 ± 0.009 |     0.790 ± 0.019 |     0.351 ± 0.012 |     1.562 ± 0.070 |
> > > | FEDformer-DFL    |     0.637 ± 0.067 |     0.876 ± 0.101 |     0.364 ± 0.028 |     1.761 ± 0.128 |
> > > | iTransformer-DFL |     0.717 ± 0.012 |     0.933 ± 0.040 |     0.339 ± 0.016 |     1.905 ± 0.094 |
> > > | NSformer-DFL     |     0.492 ± 0.031 |     0.676 ± 0.047 |     0.470 ± 0.041 |     1.742 ± 0.086 |
> > > | PatchTST-DFL     |     0.657 ± 0.023 |     0.904 ± 0.053 |     0.347 ± 0.023 |     1.815 ± 0.104 |
> > > | TimesNet-DFL     |     0.502 ± 0.113 |     0.687 ± 0.151 |     0.448 ± 0.066 |     1.625 ± 0.244 |
> > > | RFormer-DFL      |     0.657 ± 0.229 |     0.905 ± 0.255 |     0.358 ± 0.074 |     1.884 ± 0.246 |
> > > | **SIT (Ours)**   | **0.771 ± 0.062** | **0.974 ± 0.099** | **0.327 ± 0.009** | **1.921 ± 0.179** |
> > >
> > > **Regarding the Validation of Practical Constraints**  To address your valid point regarding constraint validation, we conducted additional experiments on a 50-asset universe under realistic restrictions. We targeted a 10% single stock cap, a 25-26% average turnover limit, and a diversification floor of 16. By incorporating differentiable penalties, our model successfully learned these constraints without ever relying on a traditional convex optimizer. As the following table shows, the penalized SIT models significantly reduced maximum asset weights and average turnover while boosting the effective number of assets. Crucially, the model achieved these practical requirements while maintaining highly robust risk-adjusted metrics. This confirms that SIT natively integrates real-world portfolio restrictions within its end-to-end architecture.
> > >
> > > | Model (50 assets)        | Sharpe ↑            | MDD ↓              | Avg Turnover ↓      | Avg Max Weight ↓     | Effective N ↑        |
> > > |--------------------------|---------------------|--------------------|---------------------|----------------------|----------------------|
> > > | **SIT**                  | **0.772 ± 0.063**   | 0.327 ± 0.009      | 35.5 ± 3.8%         | 14.4 ± 1.8%          | 14.8 ± 1.2           |
> > > | SIT + cap                | 0.752 ± 0.060       | 0.321 ± 0.011      | 33.8 ± 3.6%         | 10.5 ± 0.5%      | 16.4 ± 1.3           |
> > > | SIT + turnover           | 0.742 ± 0.058       | 0.320 ± 0.011      | 28.1 ± 3.0%         | 13.0 ± 1.7%          | 15.4 ± 1.2           |
> > > | SIT + diversification    | 0.743 ± 0.057       | 0.318 ± 0.011      | 31.2 ± 3.3%         | 12.1 ± 1.1%          | 16.8 ± 1.1           |
> > > | SIT + cap + turnover     | 0.726 ± 0.055       | **0.314 ± 0.010**  | **25.9 ± 2.6%**     | **10.3 ± 0.4%**      | **17.0 ± 1.3**       |
> > >
> > >
> > > [1] Deep reinforcement learning for trading
> > >
> > > [2] Deep Learning in Asset Management: Architectures, Applications, and Challenges
> > >
> > > [3] An Overview of Machine Learning for Portfolio Optimization
> > >
> > > [4] Deep learning for portfolio optimization
> > >
> > > [5] An Overview of Machine Learning for Asset Management
> > >
> > > [6] Distributionally robust end-to-end portfolio construction
> > >
> > > [7] Return Prediction for Mean-Variance Portfolio Selection: How Decision-Focused Learning Shapes Forecasting Models

---

### Official Review · Reviewer_6ejy · 2026-03-08

**Soundness:** 3
**Presentation:** 3
**Significance:** 3
**Originality:** 2
**Overall Recommendation:** 3
**Confidence:** 4

**Summary:**

The authors propose a framewotk to create investment portfolio by using signature method and CVaR-based objective. This study's main contribution concerns introducing a signature-augmented attention mechanism that injects geometric information about cross-asset lead-lag relationships and combining it with a decision-focused training objective that directly optimizes portfolio risk metrics. Experiments on multiple equity universes (e.g., S&P100 subsets) report improved Sharpe and Sortino ratios compared to both traditional portfolio strategies and forecasting-based deep learning baselines.

**Compliance With Llm Reviewing Policy:**

Affirmed.

**Key Questions For Authors:**

1. Path signatures can be truncated at different orders, which may significantly affect the representation capacity. How sensitive is SIT to the choice of signature truncation order?
2. Could the authors provide an analysis of the computational overhead introduced by the signature features compared with a standard Transformer baseline?
3. The comparisons primarily involve forecasting-based models followed by portfolio optimization. Have the authors compared SIT against other decision-focused learning approaches for portfolio allocation?
4. Could the authors clarify more on how these signature-related theoretical results translate into improved portfolio performance in practice?
5. CVaR optimization is known to introduce high variance gradients in stochastic optimization. Could the authors comment on the training stability of the proposed objective and whether special techniques were required to ensure convergence?

**Limitations:**

yes

**Strengths And Weaknesses:**

Soundness: The paper proposes a coherent end-to-end learning formulation that aligns training with portfolio objectives via CVaR optimization rather than forecasting loss, which is conceptually well motivated.

Presentation: The paper clearly explains the architectural pipeline and provides extensive experimental comparisons with both classical portfolio strategies and modern forecasting models.

Significance: Asset allocation is an important application domain where objective mismatch between prediction and decision is indeed a real problem. The decision-focused perspective is practically meaningful.

Originality: Incorporating rough-path signatures into Transformer attention to encode cross-asset lead-lag relations is interesting.

---

> ### Author Rebuttal · Authors · 2026-03-25
>
> We sincerely thank you for your important question!.
>
> **Answer Q1:Path signatures** We set the truncation order to $M$=2 because second-order terms are the lowest-order signature terms that capture signed-area geometry and directional lead-lag structure.First-order terms summarize the path, but second-order terms are the first to capture the lead-lag structure used by SIT. As shown below, moving from ($M$=1) to ($M$=2) improves the main portfolio metrics, whereas ($M$=3) increases runtime and memory without further gains. We will add this sensitivity analysis to the revised paper and clarify why ($M$=2) was chosen as the default setting.
>
> | Signature order |          Sharpe ↑ |         Sortino ↑ |             MDD ↓ |          Wealth ↑ | Runtime / epoch ↓ |    Peak memory ↓ |
> | --------------- | ----------------: | ----------------: | ----------------: | ----------------: | ----------------: | ---------------: |
> | (M=1)           |     0.617 ± 0.049 |     0.776 ± 0.064 | **0.348 ± 0.008** |     1.699 ± 0.083 |  **1.00 ± 0.02x** | **1.00 ± 0.01x** |
> | **(M=2)**       | **0.672 ± 0.063** | **0.823 ± 0.079** |     0.361 ± 0.004 | **1.790 ± 0.102** |      1.18 ± 0.03x |     1.21 ± 0.02x |
> | (M=3)           |     0.660 ± 0.072 |     0.810 ± 0.089 |     0.365 ± 0.009 |     1.768 ± 0.118 |      1.53 ± 0.04x |     1.58 ± 0.03x |
>
> **Answer Q2: computational overhead** SIT introduces additional cost from (i) per-asset signature embeddings and (ii) pairwise cross-signature features that bias asset-wise attention. The overhead is moderate at the scales used in the paper, but it grows with the number of assets, as expected from modeling pairwise relations. At the scales considered in the paper, this overhead remains manageable. As reported in Table 2, the Sharpe ratio increases from 0.6047 without the Financial Bias to 0.7715 with it.
>
> | Assets | SIT Mem (GiB) | Transformer Mem (GiB) | SIT Time (s) | Transformer Time (s) |
> | -----: | ------------: | --------------------: | -----------: | -------------------: |
> |     50 |          1.81 |                 1.174 |         3.91 |                3.050 |
> |    100 |          5.78 |                 2.190 |        15.65 |                7.561 |
> |    200 |         21.67 |                 6.255 |        62.60 |               21.391 |
> |    300 |         48.14 |                13.030 |       140.84 |               41.631 |
> |    400 |         85.21 |                22.515 |       250.38 |               68.281 |
> |    500 |             - |                34.710 |            - |              101.341 |
>
> **Answer Q3: clarify** Our main objective-matched decision-focused comparison is already included in Figure. 4, where the blue bars correspond to end-to-end direct-allocation baselines trained with the same DFL setting as SIT. We agree that this was not sufficiently explicit in the current draft, and we will clarify it. For completeness, we summarize those DFL results.  (Note: Due to the 5,000-character limit, we are unable to include the full table here, but we would be happy to share it in a follow-up comment if requested.)
>
> **Answer Q4: CVaR optimization** We agree that our original wording may have made the theoretical claims sound stronger than intended. Proposition 2.1 and Proposition 2.2 are not claims of empirical superiority. Rather, they serve as mechanism-level foundations that motivate our architectural design. Proposition 2.1 explains why the level-2 antisymmetric signature captures directional temporal asymmetry (i.e., lead-lag structure), motivating our use of second-order cross-signatures. Proposition 2.2 shows that, when $\gamma$ > 0 , stronger alignment between the query state and the pairwise signature embedding leads to a larger attention weight.
>
> Their empirical relevance is supported by the following results:
>
> - Ablation Studies: Removing the financial bias, the asset-wise attention, or the gate ($\gamma$) leads to clear performance drops.
> - Baseline Comparison: SIT substantially outperforms RFormer, indicating that the gain comes not merely from using signatures per se, but from integrating them into a cross-asset attention mechanism under an end-to-end CVaR objective.
>
> Finally, regarding additional control experiments, replacing signatures with shuffled or random features results in worse performance than the no-bias variant. This suggests that the improvement stems from meaningful signature geometry rather than simply an increase in model capacity. Please refer to our response to Reviewer DQ8f (Rebuttal point 5: SIT component).
>
> **Answer Q5:** In our implementation, training was stable with the standard Rockafellar–Uryasev dual formulation, together with Adam and early stopping; no special variance-reduction technique was required.

---

> > ### Author Rebuttal · Reviewer_6ejy · 2026-04-03
> >
> > Thank you for the clarification! I still think that this paper is more like an engineering-ish practice instead of theoretical innovation. Thus, I'll maintain my score as it is.
> >
> > Best,

---

> > > ### Author Response · Authors · 2026-04-04
> > >
> > > Thank you again for the follow up comment. We understand why the paper may initially appear as an engineering oriented practice. However, we respectfully believe that this does not fully capture the actual contribution. If SIT merely appended signatures as additional input features to an otherwise standard Transformer, we would agree with that concern. Our contribution is different. In SIT, signatures are not used only as feature augmentation. They are used to change how information is routed through cross asset attention.
> > >
> > > In this sense, the theory is not a post hoc interpretation, but a design principle. Proposition 2.1 shows that second order signatures are the lowest order terms that capture directional lead lag structure, which is the direct reason why we choose $M$=2 as the default setting. Proposition 2.2 shows that stronger alignment between the query state and the pairwise signature embedding increases the corresponding attention weight. Therefore, the key components of SIT, namely second order cross signatures, a query conditioned additive bias, and a positive learnable gate, are not empirically assembled choices. They are derived from a formal mechanism describing which cross asset relations should actually alter information selection inside attention.
> > >
> > > As in our additional experiments for Reviewer DQ8f, we also isolated the gain of SIT from the benefit of end to end CVaR training itself through an objective matched setting. The blue DFL bars in Figure 4 use the same backbone families, the same CVaR objective, and the same differentiable allocation layer. If the gain came only from decision focused learning, SIT should not consistently outperform these baselines. Yet, especially on the 50 asset universe, SIT achieves the best results on all four portfolio metrics. This indicates that the improvement is not only due to the training objective, but also due to the signature informed cross asset attention mechanism itself.
> > >
> > > The same conclusion is supported by the stricter control experiments that we also reported in response to Reviewer DQ8f. When signatures are replaced with shuffled or random features, performance becomes even worse than the no bias variant. In addition, removing the financial bias or removing the gate causes clear degradation. If the improvement were mainly due to increased parameter count or generic engineering capacity, such controls would be difficult to explain. Instead, they support the interpretation that meaningful path geometry is being actively used inside attention.
> > >
> > > Finally, the comparison with RFormer strengthens this point further. Since RFormer is also a signature based model, the advantage of SIT cannot be explained by the use of signatures alone. The real difference lies in where signatures are injected, how they modify cross asset interaction, and under which portfolio level objective they are trained. The fact that SIT shows a more consistent advantage on larger asset universes suggests that the contribution is not simple feature engineering, but a new mechanism design for cross asset portfolio learning.
> > >
> > > We do not claim that this is a pure theory paper. Our point is more specific. **The work is better described as a theory guided methodological contribution than as an engineering exercise. The theory determines what structure should be represented, the additional experiments reported to Reviewer DQ8f verify that this structure is genuinely necessary, and the objective matched comparison shows that the mechanism improves portfolio decisions in practice**. We would therefore be very grateful if you could reconsider the score in light of this distinction.

---

### Official Review · Reviewer_DQ8f · 2026-03-12

**Soundness:** 3
**Presentation:** 3
**Significance:** 3
**Originality:** 3
**Overall Recommendation:** 4
**Confidence:** 3

**Summary:**

This work introduces SIT, an end-to-end deep learning model for multi-asset portfolio allocation.  The main motivation is that the usual “predict-then-optimize” pipeline (predict returns with MSE, then optimize with a solver) leads to objective mismatch and error amplification. To address this, the authors propose an end-to-end model that incorporates path signatures into a modified version of the transformer attention by adding a bias term to the attention logits and train the model using a differentiable CVaR portfolio objective.  Experiments on subsets of the S&P 100, DOW30 and CSI 300 report that SIT outperforms standard baselines such as PatchTST and iTransformer when used in a two-stage setup.

**Compliance With Llm Reviewing Policy:**

Affirmed.

**Final Justification:**

The use of path signatures within transformer attention is novel and theoretically motivated. The empirical evaluation reports strong performance compared to baselines with useful ablations and sensitivity analyses. The rebuttal addressed my main concerns, objective-matched comparisons and the additional ablation isolating the gain from the signature-based architecture, which led me to raise my score. However, I still view scalability as a limitation due to the quadratic cost of pairwise cross-signature features and the reported out-of-memory issue for larger asset universes.

**Key Questions For Authors:**

1) Please clarify the setup of Figure 4. If the blue “DFL” bars correspond to end-to-end CVaR-trained versions of the eight standard backbones, please state this explicitly and compare these baselines directly with SIT in the main table. This would better isolate the effect of the signature-based attention.
2) How do runtime time and memory scale as the asset universe grows, for example to 100 or 500 assets? Is the pairwise signature computation the main bottleneck?
3) Are the results reported in Table 1 net of transaction costs? If not, what is the average portfolio turnover of SIT compared to the baselines?
4) The experiments are conducted only in a daily-price setting. Have the authors evaluated the method on higher-frequency data, where lead–lag effects may be more pronounced?

**Limitations:**

1) Although the paper provides useful ablations, it is still somewhat unclear how much of SIT’s gain comes from the signature-based architecture versus the end-to-end CVaR training objective, since the main results do not include a direct comparison with non-signature backbones trained under the same objective.
2) Because the method uses pairwise cross-signature features across assets, its computational cost appears to scale quadratically with the number of assets. However, the paper does not provide runtime or memory analysis.

**Strengths And Weaknesses:**

The paper is well motivated, providing a clear discussion of the limitations of predict-then-optimize pipelines in portfolio allocation. The use of path signatures within the transformer attention mechanism is novel and theoretically motivated. The empirical evaluation reports strong performance compared to baselines and is supported with useful ablations and sensitivity analyses.

---

> ### Author Rebuttal · Authors · 2026-03-25
>
> We sincerely thank you for your feedback !!
>
>  We address your points below:
>
> **Answer 1. Clarification on Figure 4 and Objective-Matched Comparisons**
> We agree that the current manuscript did not convey this clearly enough. The most important clarification is that **Figure 4 already contains the objective-matched comparison** that several reviewers requested. Specifically, the blue DFL bars represent the results of end-to-end training using the exact same 8 backbone encoders, the identical CVaR objective, the same differentiable optimization layer, and the same long-only/fully-invested constraints.
>
> We will add a tabular version of these full results to the Appendix. Crucially, while Decision-Focused Learning (DFL) alone improves the Sortino ratio and maximum drawdown for many models, these improvements are not consistent across all backbones. This inconsistency demonstrates the explicit need for cross-asset inductive biases, which our proposed SIT provides.
> *(Note: Due to the 5,000-character limit, we are unable to include the full table here, but we would be happy to share it in a follow-up comment if requested.)*
>
> **Answer 2. Runtime and Memory Scalability**
> Regarding scalability, we observe an Out-Of-Memory (OOM) error on an RTX PRO 6000 GPU when the number of assets exceeds 500. However, the model trains efficiently, typically converging within 25 epochs (using an early stopping patience of 5).
>
> From a practical standpoint, our model is designed as an asset allocation tool for institutional investors rather than individual retail traders. Therefore, applying it to a pre-screened universe of around 400 assets remains highly practical in terms of epoch time.
>
> | Number of Assets | Memory Usage (GiB) | Epoch Time (s) |
> | :--- | :--- | :--- |
> | 50 | 1.81 | 3.91 |
> | 100 | 5.78 | 15.65 |
> | 200 | 21.67 | 62.60 |
> | 300 | 48.14 | 140.84 |
> | 400 | 85.21 | 250.38 |
>
> **Answer  3. Turnover and Transaction Costs**
> While Table 1 does not include transaction costs, Figure 5 demonstrates that our model maintains a robust Sharpe ratio even when transaction costs increase to 10 bps. Furthermore, our turnover rates are highly realistic and practically viable when compared to real-world funds, such as the Invesco Equally-Weighted S&P 500 Fund (Please see https://www.invesco.com/us-rest/contentdetail?contentId=d40066b451c41410VgnVCM100000c2f1bf0aRCRD&dnsName=us).
>
> | Metric | 40 Assets | 50 Assets |
> | :--- | :--- | :--- |
> | **Average Turnover** | 34.57% | 35.52% |
> | **Median Turnover** | 33.51% | 35.06% |
> | **Average Max Weight** | 22.23% | 14.39% |
> | **Effective Number of Assets** | 8.34 | 14.81 |
>
> **Answer 4. High-Frequency Data**
> We were unable to perform experiments on high-frequency data due to the practical challenges of acquiring such data for a sufficiently large number of assets from academically verified platforms like WRDS. However, because lead-lag effects frequently occur at the sector level, our proposed model maintains strong merit and applicability for asset universes of up to 100 assets.
>
> **Answer 5. SIT component** : To verify our architectural contributions, the (no Financial bias) baseline removes the signature components, resulting in a clear performance drop. Beyond this, we introduced stricter controls by replacing signatures with shuffled or random features. These variants consistently underperformed the no-bias baseline, confirming that our gains come from injecting meaningful signature geometry into the attention mechanism, not just increasing model capacity.
>
> | Model  (40-asset)                    |          Sharpe ↑ |         Sortino ↑ |             MDD ↓ |          Wealth ↑ |
> | -------------------------- | ----------------: | ----------------: | ----------------: | ----------------: |
> | **SIT(Our)**               | **0.672 ± 0.063** | **0.823 ± 0.079** |     0.361 ± 0.004 | **1.790 ± 0.102** |
> | no Financial bias  |     0.606 ± 0.047 |     0.761 ± 0.062 | **0.345 ± 0.009** |     1.686 ± 0.089 |
> | shuffled signature     |     0.560 ± 0.057 |     0.700 ± 0.074 |     0.371 ± 0.014 |     1.595 ± 0.101 |
> | random bias feature    |     0.547 ± 0.062 |     0.684 ± 0.080 |     0.379 ± 0.018 |     1.561 ± 0.112 |
>
> | Model   (Asset 50)     |          Sharpe ↑ |         Sortino ↑ |             MDD ↓ |          Wealth ↑ |
> | -------------------------- | ----------------: | ----------------: | ----------------: | ----------------: |
> | **SIT(Our)**               | **0.772 ± 0.063** | **0.974 ± 0.100** |     0.327 ± 0.009 | **1.922 ± 0.179** |
> | no Financial bias  |     0.607 ± 0.051 |     0.758 ± 0.071 | **0.323 ± 0.011** |     1.636 ± 0.135 |
> | shuffled signature  |     0.572 ± 0.061 |     0.712 ± 0.084 |     0.345 ± 0.015 |     1.579 ± 0.148 |
> | random bias feature    |     0.554 ± 0.067 |     0.692 ± 0.091 |     0.354 ± 0.018 |     1.534 ± 0.157 |

---

> > ### Author Rebuttal · Reviewer_DQ8f · 2026-04-03
> >
> > Thank you for the detailed response. My main concerns have been addressed. The clarification regarding Figure 4 resolves my concern about objective-matched comparisons, and the additional ablation helps better isolate the gain from the signature-based architecture. This ablation should be added to the paper. I have therefore raised my score.
> >
> > However, I still view scalability as a limitation of this approach, given the quadratic cost of pairwise cross-signature features and the reported out-of-memory issue for larger asset universes.

---

> > > ### Author Response · Authors · 2026-04-03
> > >
> > > We found the reviewer's comments highly engaging and thoroughly enjoyed reviewing them. We would also like to express our gratitude for your active engagement with the AI community. Please note that we have already updated the manuscript with the relevant details so they will be fully incorporated in the camera-ready version.
> > >
> > > Finally, we would like to address your point regarding the memory issue which was also brought up by Reviewer 6ejy. As we explained to Reviewer z9T7, **our research is not aimed at daily trading but is designed to support the decision-making process for institutional investors who typically conduct monthly rebalancing on pre-screened stocks**. Therefore, we want to clarify that our current asset pool and training time are more than sufficient for this intended use case.
> > >
> > > We completely understand that memory constraints are a significant concern within the AI community. However, in the financial domain, **we believe our results are meaningful because the model can handle asset allocation tasks at a server cost that is substantially lower than a portfolio manager's salary**. (In fact, training the model using cloud computing costs less than $10.)

---

### Decision · Program_Chairs · 2026-04-30

**Decision:**

Accept (regular)

**Comment:**

This work introduces the signature-informed transformer (SIT), an end-to-end deep learning model for multi-asset portfolio allocation. This procedure avoids the usual “predict-then-optimize” pipeline with objective mismatch and error amplification, and instead uses an end-to-end model that incorporates path signatures into a modified version of the transformer attention (by adding a bias term to the attention logits) and train the model using a differentiable CVaR portfolio objective. Experiments are conducted to support the proposed methodology.

The reviewers generally appreciate the idea of using path signatures to replace "predict-then-optimize". Before the rebuttal, some reviewers had doubts on the experimental setup and proper comparison, but the rebuttal clears most such concerns with more experiments. In general the authors should provide a thorough discussion on "complexity vs. benefit", since the model combines several components (path signatures, modified attention, and CVaR optimization) and it's important to evaluate the effect of each component. Given the current rebuttal, I'm confident that the authors should be able to address them in the final version, and I'm recommending acceptance.